# 🎲 GS2E: Gaussian Splatting is an Effective Data Generator for Event Stream Generation

Yuchen Li[1,*]    Chaoran Feng[1,*]    Zhenyu Tang[1]    Kaiyuan Deng[2]    Wangbo Yu[1]
Yonghong Tian[1,†]    Li Yuan[1,†]

[1]Peking University, Shenzhen Graduate School
[2]University of Arizona, Department of Electrical and Computer Engineering
{yuchenli, chaoran.feng, zhenyutang, wbyu}@stu.pku.edu.cn, {yhtian, yuanli-ece}@pku.edu.cn

https://intothemild.github.io/GS2E.github.io/

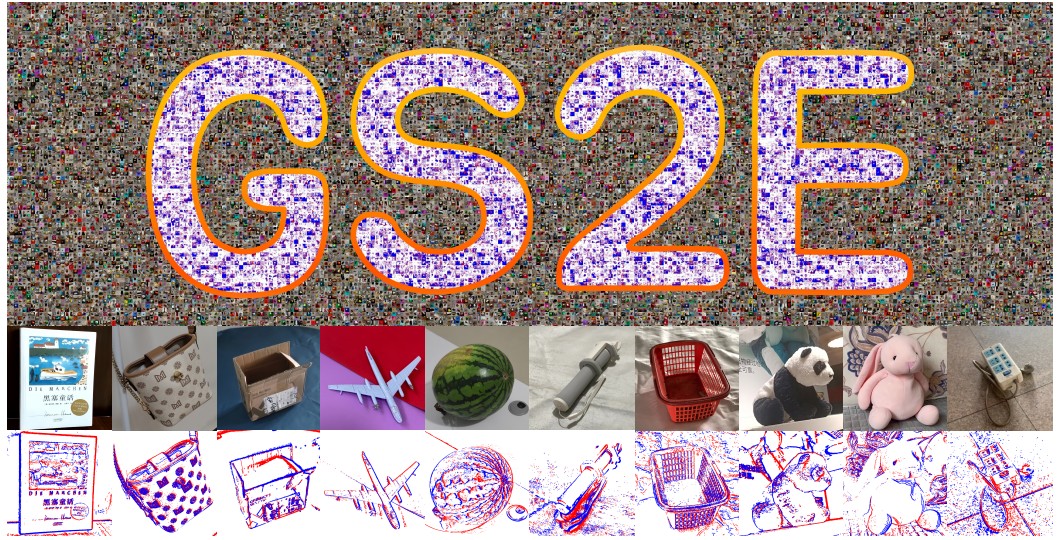

Figure 1: We propose **GS2E**, a high-fidelity synthetic dataset designed for 3D event-based vision, comprising over **1150** scenes. GS2EExamples of RGB frames and event streams are shown above.

## Abstract

We introduce (GS2E) (GAUSSIAN SPLATTING TO EVENT GENERATION), a large-scale synthetic event dataset for high-fidelity event vision tasks, captured from real-world sparse multi-view RGB images. Existing event datasets are often synthesized from dense RGB videos, which typically lack viewpoint diversity and geometric consistency, or depend on expensive, difficult-to-scale hardware setups. GS2E overcomes these limitations by first reconstructing photorealistic static scenes using 3D Gaussian Splatting, and subsequently employing a novel, physically-informed event simulation pipeline. This pipeline generally integrates adaptive trajectory interpolation with physically-consistent event contrast threshold modeling. Such an approach yields temporally dense and geometrically consistent event streams under diverse motion and lighting conditions, while ensuring strong alignment with underlying scene structures. Experimental results on event-based 3D reconstruction demonstrate GS2E's superior generalization capabilities and its practical value as a benchmark for advancing event vision research.

---

*These authors contributed equally to this work.

†Corresponding author.

39th Conference on Neural Information Processing Systems (NeurIPS 2025) Track on Datasets and Benchmarks.

# 1  Introduction

Event cameras, provide high temporal resolution, low latency, and high dynamic range, making them uniquely suited for tasks involving fast motion and challenging lighting conditions [10, 74]. These advantages have been demonstrated in various applications such as autonomous driving [15, 17], drone navigation [56, 5], and 3D scene reconstruction [25, 73, 24, 57, 52, 72]. In particular, their ability to capture asynchronous brightness changes enables accurate motion perception for 3D reconstruction and novel view synthesis (NVS) tasks, surpassing the capabilities of conventional RGB sensors under fast motion and dynamic illumination [75]. However, despite their potential, the advancement of event-based vision algorithms is significantly limited by the scarcity of large-scale, high-quality event datasets, especially those offering multi-view consistency and aligned RGB data. This bottleneck has slowed the development of hybrid approaches that aim to combine event and RGB signals for high-fidelity 3D scene understanding and reconstruction. While event streams provide accurate geometric and motion cues through the high-frequency edge information, RGB frames, often motion-blurred, contribute essential color features with low-frequent details but can suffer from degraded textural information for rendering. While event streams offer precise geometric and motion cues through high-frequency edge information, RGB frames, though often degraded by motion blur, provide complementary low-frequency texture and essential color details for photorealistic rendering. However, the lack of large-scale datasets that jointly exploit these complementary signals limits progress in event-based 3D scene understanding and generation [76]. This aligns with the broader trend of establishing structured evaluation paradigms across domains [34].

As illustrated in Figure 2, existing efforts to conduct event-based 3D reconstruction datasets fall into three main categories: (1) *Real-world capture:* This involves dedicated hardware setups such as synchronized event-RGB stereo rigs or multi-sensor arrays (e.g., DAVIS-based systems [64]). While providing realistic data, these setups are expensive, prone to calibration errors, and difficult to scale to diverse scenes and camera configurations, as seen in systems like Dynamic EventNeRF [58]. (2) *Video-driven synthesis:* v2e [23] and Vid2E [12] generate event streams from dense, high-framerate RGB videos. Although flexible and accessible, they suffer from limited viewpoint diversity and lack geometric consistency, making them suboptimal for multi-view reconstruction tasks. (3) *Simulation via computer graphics engines:* Recent approaches [54, 19, 27, 41] leverage 3D modeling tools like Blender [6] or Unreal Engine [7] to simulate photo-realistic scenes and generate event data along with RGB, depth, and pose annotations. These allow fine-grained control over camera trajectories and lighting, enabling multi-view, physically-consistent dataset synthesis. However, such pipelines may introduce a domain gap due to non-photorealistic rendering or oversimplified dynamics [61].

Video-driven and graphics-based approaches offer greater scalability by enabling controllable and repeatable data generation among above methods. However, video-based methods rely on densely sampled RGB frames from narrow-baseline views, often resulting in motion blur and limited geometric diversity. Simulation with physical engines offers greater control over scenes and trajectories, yet frequently suffer from domain gaps due to non-photorealistic rendering and simplified dynamics [50]. Moreover, an important yet often underexplored factor influencing the realism of synthetic event data is the contrast threshold (CT), which defines the minimum log-intensity change required to trigger an event. While many existing simulators [37, 12] adopt fixed or heuristic CT values, recent analyses [61, 26, 18] show that CT values vary considerably across sensors, scenes, and even within the same sequence. This variability induces a significant distribution shift between synthetic and real event data, thereby limiting the generalization capability of models trained on simulated streams. We posit that accurate modeling of contrast thresholds as data-dependent and adaptive parameters is crucial for generating realistic and transferable event representations. Incorporating physically informed CT sampling, potentially complemented by plausible noise considerations, can significantly enhance sim-to-real transferability in downstream tasks such as 3D reconstruction [79, 9, 78, 9, 86] and optical flow estimation [90, 47, 8, 33, 11].

Based on the above observation, we propose a novel pipeline for synthesizing high-quality, geometry-consistent multi-view event data from sparse RGB inputs. Leveraging 3D Gaussian Splatting (3DGS) [31], we first reconstruct photorealistic 3D scenes from a sparse set of multi-view images with known poses. We then generate continuous trajectories via adaptive interpolation and render dense RGB sequences along these paths. These sequences are fed into our physically-informed event simulator [37]. This simulator employs our data-driven contrast threshold modeling to ensure event responses are consistent with real sensor behaviors, and inherently maintains geometric consistency

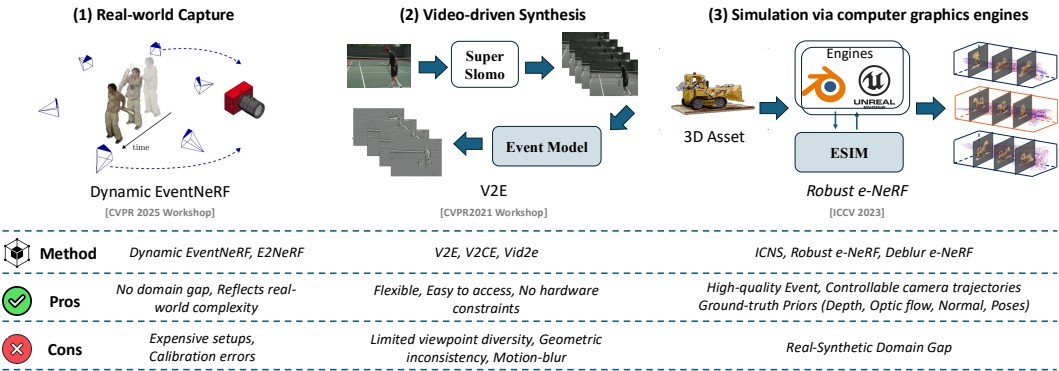

| | Method | Dynamic EventNeRF, E2NeRF | V2E, V2CE, Vid2e | ICNS, Robust e-NeRF, Deblur e-NeRF |
|---|---|---|---|---|
| | Pros | No domain gap, Reflects real-world complexity | Flexible, Easy to access, No hardware constraints | High-quality Event, Controllable camera trajectories Ground-truth Priors (Depth, Optic flow, Normal, Poses) |
| | Cons | Expensive setups, Calibration errors | Limited viewpoint diversity, Geometric inconsistency, Motion-blur | Real-Synthetic Domain Gap |

Figure 2: **Overview and comparison of event-based 3D dataset construction methods.** We compare *(1) real-world capture*, *(2) video-driven synthesis*, and *(3) simulation via computer graphics engines* in terms of commonly used methods, strengths, and drawbacks.

through the 3DGS-rendered views and trajectories. Our approach requires no dense input video and preserves the geometric fidelity of the original scene. The controllable virtual setup enables diverse motion patterns and blur levels, supporting the training of robust event-based models.

To summarize, our main contributions are:

- We propose a novel simulation pipeline for generating multi-view event data from sparse RGB images, leveraging 3DGS for high-fidelity reconstruction and novel view synthesis.
- We propose an adaptive trajectory interpolation strategy coupled with a physically-grounded contrast threshold model, jointly enabling the synthesis of temporally coherent and sensor-consistent event streams.
- We construct and release a benchmark dataset comprising photorealistic RGB frames, motion-blurred sequences, accurate camera poses, and multi-view event streams, facilitating research in structure-aware event vision.

## 2 Related Work

### 2.1 Optimization-based Event Simulators

Early event simulation methods, such as those proposed by Kaiser et al.[29] and Mueggler et al.[45], generated events by thresholding frame differences or rendering high-framerate videos. However, these approaches failed to capture the inherent asynchronous and low-latency characteristics of event sensors. Subsequent works like ESIM [54] and Vid2E [13] improved realism by incorporating per-pixel log-intensity integration and optical flow-based interpolation to approximate event triggering more faithfully. V2E [23] further advanced realism by modeling sensor-level attributes such as bandwidth limitations, background activity noise, and Poisson-distributed firing. More recent simulators including V2CE [84], ICNS [28], and DVS-Voltmeter [37], introduced hardware-aware components, accounting for effects such as latency, temperature-dependent noise, and local motion dynamics. PECS [20] extended this direction by modeling the full optical path through multispectral photon rendering. Despite their increased physical fidelity, most of these simulators operate purely on 2D image or video inputs and do not exploit the underlying 3D structure of scenes. Furthermore, the prevalent use of fixed contrast thresholds across all viewpoints and scenes fails to reflect the variability observed in real sensors, thereby introducing a significant domain gap in simulated data.

### 2.2 Learning-based Event Simulators

Recent efforts have explored deep learning to synthesize event streams in a data-driven manner. EventGAN [89] employed GANs to generate event frames from static images, while Pantho et al. [48] learned to generate temporally consistent event tensors or voxel representations. Domain-adaptive simulators such as Gu et al. [16] jointly synthesized camera trajectories and event data, improving realism under target distributions. However, learning-based approaches generally suffer

from limited interpretability and require retraining when transferred to new scenarios, leading to weaker robustness compared to physics-inspired models. In contrast, our method follows a physically grounded yet geometry-aware paradigm: we first reconstruct high-fidelity 3D scenes via 3DGS then synthesize event streams by simulating photorealistic motion blur and modeling the contrast threshold distribution observed in real-world data. This enables us to generate temporally coherent, multi-view consistent event data with improved realism and transferability.

## 3 Method

### 3.1 Preliminary

**3D Gaussian Splatting.** 3D Gaussian Splatting [31] represents a scene as a set of anisotropic Gaussians. Each Gaussian $G_i$ is defined by

$$G_i(\mathbf{x}) = \exp\left(-\tfrac{1}{2}\,(\mathbf{x} - \boldsymbol{\mu}_i)^\top (\mathbf{R}_i \, \mathrm{diag}(\mathbf{s}_i)^2 \, \mathbf{R}_i^\top)^{-1}(\mathbf{x} - \boldsymbol{\mu}_i)\right), \tag{1}$$

where $\boldsymbol{\mu}_i \in \mathbb{R}^3$ is the mean, $\mathbf{q}_i \mapsto \mathbf{R}_i \in \mathrm{SO}(3)$ is the rotation, and $\mathbf{s}_i \in \mathbb{R}_+^3$ is the scale. View-dependent radiance coefficients $\mathbf{c}_i$ and opacity $\alpha_i \in [0, 1]$ are optimized via differentiable rasterization under an $\ell_1$ photometric loss. Each Gaussian is transformed by the camera pose $\mathbf{T} \in \mathrm{SE}(3)$ and projected at render time, then the resulting 2D covariance is

$$\boldsymbol{\Sigma}_i' = \mathbf{J}_i \, \mathbf{T} \, \boldsymbol{\Sigma}_i \, \mathbf{T}^\top \, \mathbf{J}_i^\top, \tag{2}$$

where $\mathbf{J}_i$ denotes the Jacobian of the projection and each pixel colors $\hat{C}$ are composited as follows:

$$\hat{C} = \sum_{k \in \mathcal{N}} \mathbf{c}_k \, \alpha_k \prod_{j < k}(1 - \alpha_j). \tag{3}$$

**Event Generation Model.** Event cameras produce an asynchronous stream of tuples $(x, y, t_i, p_i)$ by thresholding changes in log-irradiance [10, 14]. Denoting the last event time at pixel $(x, y)$ by $t_{\mathrm{ref}}$, define as:

$$\Delta \log L = \log L_{(x,y)}(t_i) - \log L_{(x,y)}(t_{\mathrm{ref}}). \tag{4}$$

An event of polarity $p_i \in \{+1, -1\}$ is emitted whenever $|\Delta \log L| \geq c$:

$$p_i = \begin{cases} +1, & \Delta \log L \geq c, \\ -1, & \Delta \log L \leq -c. \end{cases} \tag{5}$$

After each event, $t_{\mathrm{ref}}$ is updated to $t_i$. This simple thresholding mechanism yields a high-temporal-resolution, sparse stream of brightness changes suitable for downstream vision tasks. Here, we introduce the off-the-shell model DVS-Voltmeter [37] as the event generation model, which incorporates physical characteristics of DVS circuits. Unlike deterministic models, DVS-Voltmeter treats the voltage evolution at each pixel as a stochastic process, specifically a Brownian motion with drift. In this formulation, the photovoltage change $\Delta V_d$ over time is modeled as

$$\Delta V_d(t) = \mu \Delta t + \sigma W(\Delta t), \tag{6}$$

where $\mu$ is a drift term capturing systematic brightness changes, $\sigma$ denotes the noise scale influenced by photon reception and leakage currents, and $W(\cdot)$ represents a standard Brownian motion. Events are then generated when the stochastic voltage process crosses either the *ON* or *OFF* thresholds. This physics-inspired modeling enables it to produce events with realistic timestamp randomness and noise characteristics, providing more faithful supervision for event-based vision tasks.

### 3.2 Pipeline Overview

Our pipeline generates multi-view, geometry-consistent event data from sparse RGB inputs. The process begins with collecting sparse multi-view RGB images along with their corresponding camera poses (§3.3). Using these inputs, we reconstruct high-fidelity scene geometry and appearance via 3DGS [31] (§3.4), providing a solid foundation for the subsequent steps. To simulate diverse observations, we generate smooth, controllable virtual camera trajectories by reparameterizing the original pose sequence based on velocity constraints, followed by interpolation of dense viewpoints along the trajectory (§3.5). Finally, the generated RGB sequences are fed into our optimized event generation module to synthesize temporally coherent, multi-view-consistent event streams (§3.7). This well-structured pipeline enables scalable and controllable event data generation from sparse RGB inputs, ensuring both accuracy and efficiency. The overall pipeline is shown in Figure 3.

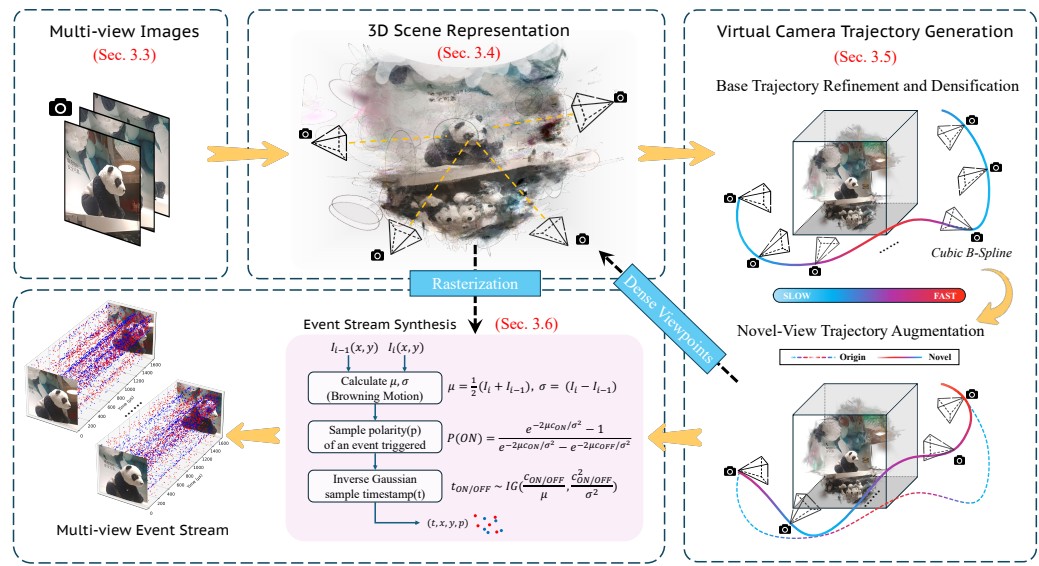

Figure 3: Overview of the proposed **GS2E** pipeline. Starting from sparse multi-view RGB images and known camera poses, we reconstruct high-fidelity scene representations using 3D Gaussian Splatting. Virtual camera trajectories are then synthesized via velocity-aware reparameterization and interpolation. The rendered image sequences are passed to a volumetric event simulator to generate temporally coherent and geometrically consistent event streams.

## 3.3 Data Collection

To support high-fidelity reconstruction and geometry-consistent event generation, we leverage two complementary datasets. The first is MVImgNet [80], a large-scale multi-view image collection comprising 6.5 million frames from 219,199 videos across 238 object categories. We sample **1,000** diverse scenes suitable for 3D reconstruction and motion-aware event synthesis from this dataset. To supplement MVImgNet's object-centric diversity with scene-level structural richness, we incorporate DL3DV [38], a dense multi-view video dataset offering accurate camera poses and ground-truth depth maps across 10,000 photorealistic indoor and ourdoor scenes. We also sample **50** diverse scenes from its 140 benchmark scenes. DL3DV provides high-quality geometry and illumination cues that are critical for evaluating spatial and temporal consistency in event simulation.

Totally, we select **1050** scenes from these datasets which enable us to construct a diverse benchmark for sparse-to-event generation, supporting both object-level and scene-level evaluation under motion blur and asynchronous observation conditions.

## 3.4 3D Scene Representation

We employ 3DGS as detailed in §3.1 to reconstruct high-fidelity 3D scenes from sparse input views. These views are represented by their corresponding camera poses $\{P_i = (R_i, \mathbf{T}_i)\}_{i=1}^{N}$, where $R_i \in \mathrm{SO}(3)$ is the rotation matrix, and $\mathbf{T}_i \in \mathbb{R}^3$ is the translation vector. For the MVImgnet and DL3DV datasets, we typically use $N = 30$ and $100$ for the number of input views. Given the image sequence $\{I_i\}_{i=1}^{N}$, we train a 3DGS model for 30,000 iterations to reconstruct a high-fidelity 3D radiance field. This radiance field captures both the scene's geometry and appearance, serving as the foundation for subsequent trajectory interpolation and event stream synthesis.

## 3.5 Virtual Camera Trajectory Generation

To simulate continuous camera motion essential for realistic event data synthesis, we transform the initial discrete set of camera poses, often obtained from structure-from-motion with COLMAP [59], into temporally dense and spatially smooth trajectories. This process involves two primary stages: (1) initial trajectory refinement and adaptive densification, (2) followed by an optional augmentation stage for enhanced motion diversity.

### 3.5.1 Base Trajectory Refinement and Densification

The raw camera poses $\{P_i = (R_i, \mathbf{T}_i)\}_{i=1}^{N}$ can exhibit jitter or abrupt transitions, detrimental to high-fidelity event simulation. We first address this through local pose smoothing and then generate a dense base trajectory using velocity-controlled interpolation.

**Pose Stabilization via Local Trajectory Smoothing.** To mitigate local jitter and discontinuities, we apply a temporal smoothing filter to the original camera poses. For each pose $P_i$, we define a local temporal window $\mathcal{W}_i = \{P_j \mid |j - i| \leq w, \ j \in N^+\}$ with a half-width $w$ (e.g., $w = 2$). The smoothed pose $P_i' = (R_i', \mathbf{T}_i')$ is computed as:

$$\mathbf{T}_i' = \frac{1}{|\mathcal{W}_i|} \sum_{j \in \mathcal{W}_i} \mathbf{T}_j, \tag{7}$$

$$R_i' = \mathrm{Slerp}\left(\{R_j\}_{j \in \mathcal{W}_i}, \frac{1}{2}\right), \tag{8}$$

where $\mathrm{Slerp}(\cdot)$ denotes spherical linear interpolation of rotations, evaluated at the temporal midpoint of the window. This procedure enhances local continuity, yielding a smoothed sequence $\{P_i'\}_{i=1}^{N}$ suitable for subsequent densification.

**Velocity-Controlled Dense Interpolation.** Building upon the smoothed poses $\{P_i'\}$, we generate a temporally uniform but spatially adaptive dense trajectory. Given a desired interpolation multiplier $\gamma > 1$, the target number of poses in the dense trajectory is $M = \lceil \gamma \cdot N \rceil$. These poses, $\{\tilde{P}_j = (\tilde{R}_j, \tilde{\mathbf{T}}_j)\}_{j=0}^{M-1}$, are sampled at evenly spaced normalized time steps $t_j = j/(M - 1)$. To achieve adaptive spatial sampling, we first quantify the motion between adjacent smoothed poses. The displacement $\delta_i$ between $P_i'$ and $P_{i+1}'$ is defined as a weighted combination of rotational and translational changes:

$$\delta_i = \alpha \cdot \theta_i + \beta \cdot \|\mathbf{T}_{i+1}' - \mathbf{T}_i'\|_2, \tag{9}$$

where $\theta_i = \cos^{-1}\left(\frac{\mathrm{Tr}(R_{i+1}'(R_i')^{\top}) - 1}{2}\right)$ is the geodesic distance between orientations $R_i'$ and $R_{i+1}'$, and $\alpha, \beta$ are weighting coefficients. The cumulative path length up to pose $P_i'$ is $s_i = \sum_{k=0}^{i-1} \delta_k$, with $s_0 = 0$. The total path length is $s_{N-1}$. We then introduce a user-defined velocity profile, which can be a continuous function $v(t)$ or a discrete list $\{v_k\}_{k=0}^{M-2}$, controlling the desired speed along the trajectory. This profile dictates the sampling density: higher velocities lead to sparser sampling in terms of path length per time step. The target path length $\tilde{s}_j$ corresponding to each time step $t_j$ is computed by normalized cumulative velocity:

$$\tilde{s}_j = s_{N-1} \cdot \frac{\sum_{k=0}^{j-1} v_k \cdot \Delta t}{\sum_{l=0}^{M-2} v_l \cdot \Delta t}, \tag{10}$$

where $\Delta t = 1/(M - 1)$. Finally, we fit a cubic B-spline curve to the control points $\{(s_i, P_i')\}$ (parameterized by cumulative path length $s_i$) and sample this spline at the reparameterized path lengths $\{\tilde{s}_j\}$ to obtain the dense trajectory $\{\tilde{P}_j\}$. This base trajectory serves as a foundation for rendering image sequences.

### 3.5.2 Novel-View Trajectory Augmentation for Enhanced Motion Diversity

To further enrich the dataset with varied camera movements, we generate multiple *novel-view mini-trajectories*. These are derived by sampling keyframes from the dense base trajectory $\{\tilde{P}_j\}_{j=0}^{M-1}$ (generated in §3.5.1) and interpolating new paths between them. Specifically, we uniformly sample $G$ groups of $K$ keyframes from $\{\tilde{P}_j\}$ without replacement:

$$\mathcal{K}_g = \left\{\tilde{P}_{i_1}^{(g)}, \ldots, \tilde{P}_{i_K}^{(g)}\right\} \subset \{\tilde{P}_j\}, \quad g = 1, \ldots, G, \tag{11}$$

where $K \leq K_{\max}$ is the number of keyframes per mini-trajectory and we set $K = 5$. For each group $\mathcal{K}_g$, we compute cumulative pose displacements along its $K$ keyframes using the metric $\delta_k$ (Eq. 9), resulting in a local path length $s_{K-1}'$. We then generate $F$ uniformly spaced spatial targets along this local path:

$$\hat{s}_\ell = \frac{\ell \cdot s_{K-1}'}{F - 1}, \quad \ell = 0, \ldots, F - 1. \tag{12}$$

Novel-view poses $\{\hat{P}_\ell^{(g)}\}_{\ell=0}^{F-1}$ are interpolated by fitting either a cubic B-spline or a Bézier curve (with a randomly selected degree $d \in \{2,3,4,5\}$) to the keyframes in $\mathcal{K}_g$, parameterized by their local cumulative path length, and then sampling at $\{\hat{s}_\ell\}$. Each interpolated pose $\hat{P}_\ell^{(g)}$ inherits its camera intrinsics from the first keyframe $\tilde{P}_{i_1}^{(g)}$ in its group. Each resulting sequence $\mathcal{C}_g = \{\hat{P}_\ell^{(g)}\}_{\ell=0}^{F-1}$ constitutes a geometry-consistent, temporally uniform mini-trajectory. The collection of all $G$ groups, $\{\mathcal{C}_g\}_{g=1}^G$, provides a diverse set of camera motions. In our experiments, we typically set $G = 3$ and $F = 150$. These augmented trajectories, along with the base trajectory, are used for rendering image sequences for event synthesis.

## 3.6 Optimize Novel View Synthesis of RGB Domain

Event synthesis is highly sensitive to minute radiometric variations and sensor noise; hence the fidelity of RGB novel view synthesis (NVS) that drives simulation is critical. Because 3DGS is a lossy reconstruction, naïve renderings may contain subtle artifacts (e.g., incompletely reconstructed fine textures) that are amplified in the event domain. We therefore explicitly optimize the RGB NVS stage.

We adopt three complementary measures to improve RGB view fidelity before event simulation: (i) *Data curation.* We use densely covered multi-view datasets (DL3DV, MVImgNet) and apply manual screening plus task-driven filtering; scenes with poor downstream performance (event-based 3D reconstruction, video reconstruction, deblurring) are removed. (ii) *RGB NVS enhancement.* A diffusion-based RGB refinement module, Difix3D+ [71], is applied on top of 3DGS to suppress holes/artifacts while preserving geometry, recovering rich textures and crisp edges without altering camera parameters. (iii) *Decoupled event noise.* RGB denoising is kept separate from event-domain stochasticity: the physically informed simulator governs event noise via sensor models like threshold distribution, preventing uncontrolled propagation of RGB artifacts.

The improvement achieved by incorporating this module is evident, with specific results detailed in the quantitative analysis of the experimental section.(§4.4)

## 3.7 Event Synthesis from Rendered Sequences

Given high-temporal-resolution image sequences rendered along diverse virtual camera trajectories (§3.5), we synthesize event streams using the DVS-Voltmeter model [37], which stochastically models pixel-level voltage accumulation. This simulator provides temporally continuous and probabilistically grounded event generation, effectively mitigating aliasing artifacts introduced by 3DGS rasterization [31], especially under fast camera motion.

As discussed in Eq. (5) (§3.1), a key parameter is the *contrast threshold* $c$, denoting the minimum log-intensity change required to trigger an event. Following the calibration strategy of Stoffregen et al. [61], we empirically sweep $c$ within $[0.25, 1.5]$, observing that:

- Low thresholds ($c \leq 0.4$) yield dense, low-noise events, resembling IJRR [46], but may introduce *floater artifacts* when used with 3DGS (see §B).
- High thresholds ($c \geq 0.8$) produce sparse events with pronounced dynamic features, similar to MVSEC [88].

The target datasets, MVImgNet [80] and DL3DV [38], generally exhibit moderate motion and textured surfaces, statistically between IJRR and HQF [61]. Based on this observation and experimental validation (§4), we adopt $c \in [0.2, 0.5]$, balancing fine detail preservation and temporal coherence through the stochastic nature of the Voltmeter model.

# 4 Experiments

**Implementation details.** For the reconstruction stage, we carefully collect approximately 2k high-quality multi-view images from 2 public datasets: MVImageNet [80] and DL3DV [38]: we choose and render 1.8k scenes from MVImageNet and 100 scenes from DL3DV. We employ the official interplementation verision of 3DGS [31] in the original setting. For the event generation stage, we utilize the DVS-Voltmeter simulator [37] to synthesize events from the rendered RGB sequences. We

adopt the following sensor-specific parameters to closely mimic the behavior of real DVS sensors: the ON and OFF contrast thresholds are both set to $\Theta_{\text{ON}} = \Theta_{\text{OFF}} = 1$ as default. We conduct our experiment on the NVIDIA RTX 3090 24GB, and more details and settings are in the appendix.

**Evaluation Baselines and Metrics.** We evaluate the proposed dataset across three key dimensions: (1) 3D reconstruction quality (§4.1), (2) the domain gap between synthetic and real-world event streams (§4.2), and (3) applicability to a range of downstream tasks (§4.3). For reconstruction-related evaluations, we adopt standard full-reference image quality metrics: PSNR, SSIM [70], and LPIPS [83], to assess both novel view synthesis and event-based deblurring performance. To further evaluate perceptual quality in image restoration and video interpolation tasks, we also employ no-reference metrics, including CLIPIQA [67], MUSIQ [30], and RANKIQA [40]. These metrics help quantify the realism, fidelity, and temporal consistency of outputs under different usage scenarios.

## 4.1 Reliability of 3D Reconstruction

We trained the 3DGS model on the MVImgNet dataset following a rigorous selection of input views and optimized the model for 30,000 iterations. We then evaluated its rendering fidelity against the ground-truth test set. Quantitative results demonstrate high reconstruction quality, with an average PSNR of 29.8, SSIM of 0.92, and a perceptual LPIPS score of 0.14. These results establish a solid foundation for leveraging 3DGS as a core module for camera pose control, high frame-rate interpolation, and photorealistic rendering, thereby enabling physically grounded event simulation.

## 4.2 Reliability of event sequences

Due to the lack of widely accepted quantitative metrics for evaluating event data quality, recent proposals such as the Event Quality Score (EQS) [3] offer promising directions for future research. However, as the EQS implementation is not publicly available.

Therefore, we first performed qualitative evaluations using real-world RGB-event datasets. Specifically, we employed the DSEC dataset from the Robotics and Perception Group at the University of Zurich, which provides synchronized recordings from RGB and DVS cameras in driving scenarios. To approximate static 3D scene conditions, we selected scenes with rigid object motion and limited amplitude variation. Visual comparisons demonstrate that our method produces event distributions more consistent with real data than conventional video-driven synthesis approaches.

Afterward, to objectively assess event-stream fidelity, we carefully reproduce the EQS on *10 real scenes* from the DSEC dataset, where ground-truth events are available. We compare video-to-event methods (Vid2e, v2e) with our GS2E, and an enhanced variant with Difix3D+.

Table 1: Evaluation of the event stream quality with the DSEC dataset (↑: higher is better).

| Metric | Vid2e | v2e | GS2E | GS2E+Difix3D+ |
|---|---|---|---|---|
| EQS↑ | 0.725 | 0.738 | **0.761** | **0.782** |

GS2E achieves higher EQS than Vid2e and v2e, and further improves with Difix3D+. We attribute these gains to: (i) scene-consistent multi-view synthesis that enables dense, artifact-reduced event generation across views; and (ii) physically informed noise modeling that captures realistic sensor behaviors without temporal jitter or stereo mismatch typical of real sensors.

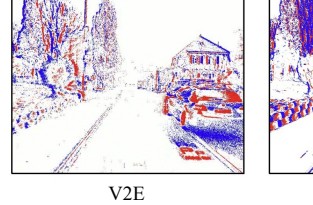 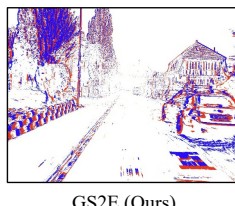 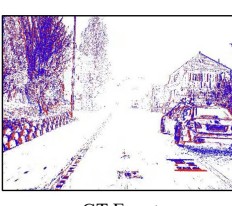 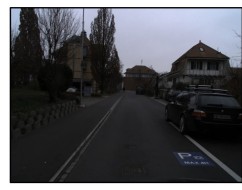

|  V2E  |  GS2E (Ours)  |  GT Event  |  GT RGB  |

Figure 4: Qualitative comparison of synthesized event distributions using GS2E versus traditional video-driven event synthesis methods, evaluated against real-world event data from the DSEC dataset.

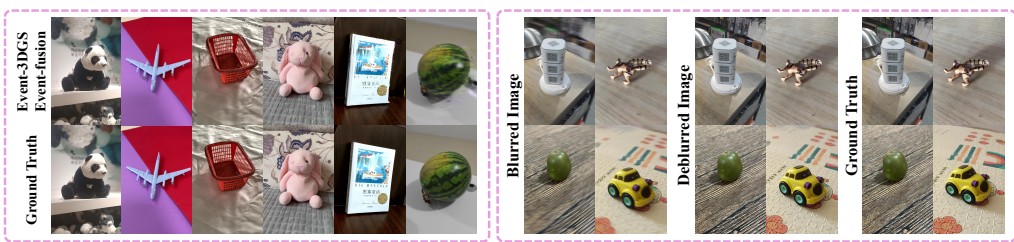

**(a) Event-vision Task: 3D Reconstruction**  **(b) Event-vision Task: Image Deblurring**

Figure 5: Application to Multiple Tasks. We benchmark it across event-vision tasks: 3D reconstruction and image deblurring.

Table 2: Comparison of different methods under varying motion speeds. Metrics are averaged over each category (↑: higher is better; ↓: lower is better).

| Category | Method | Mild Speed | | | Medium Speed | | | Strong Speed | | |
|---|---|---|---|---|---|---|---|---|---|---|
| | | PSNR↑ | SSIM↑ | LPIPS↓ | PSNR↑ | SSIM↑ | LPIPS↓ | PSNR↑ | SSIM↑ | LPIPS↓ |
| Event-only | E-NeRF [32] | 21.67 | 0.827 | 0.216 | 20.93 | 0.815 | 0.244 | 20.11 | 0.792 | 0.260 |
| | Event-3DGS [18] | 11.19 | 0.623 | 0.649 | 10.34 | 0.387 | 0.695 | 10.68 | 0.374 | 0.712 |
| Event-fusion | E-NeRF [32] | 21.89 | 0.834 | 0.208 | 21.05 | 0.820 | 0.239 | 20.57 | 0.809 | 0.251 |
| | Event-3DGS [18] | 24.31 | 0.884 | 0.118 | 21.88 | 0.832 | 0.224 | 19.36 | 0.793 | 0.295 |

## 4.3 Application to Multiple Tasks

To evaluate the generalization and practicality of our proposed event dataset, we benchmark it across three event-vision tasks: 3D reconstruction, image deblurring, and image/video reconstruction. For all tasks, we compare several state-of-the-art methods, and report standard image quality metrics.

**Event-based 3D Reconstruction**. We first evaluate the utility of our dataset in static 3D scene reconstruction. To assess robustness under motion-induced challenges, we simulate static scenes with varying camera motion speeds (*mild, medium*, and *strong*), allowing controlled evaluation of temporal consistency and appearance fidelity. We further compare grayscale-only and RGB-colored supervision settings to investigate the effect of color information. As shown in Table 2, while all methods exhibit performance drops under faster motion, models trained with color supervision consistently achieve better perceptual quality. These results highlight the versatility of our dataset in supporting both grayscale and color-aware pipelines, and its suitability for evaluating spatiotemporal consistency in static scenes captured via asynchronous event observations.

**Event-based Image Deblurring.** We further test whether event streams generated from our dataset can support high-quality image restoration under motion blur. Leveraging recent event-guided deblurring frameworks [60, 62], we evaluate reconstruction performance using both synthetic blurry frames paired with our event data. The results indicate that our dataset effectively captures motion-dependent blur patterns and high-frequency temporal cues, which help deblurring models produce sharper and more temporally consistent outputs, particularly in low-light and fast-moving scenes.

Table 3: Comparison of image deblurring and video reconstruction methods.

| Method | PSNR↑ | SSIM↑ | LPIPS↓ |
|---|---|---|---|
| *Deblurring Task* | | | |
| D2Net [60] | 29.61 | 0.932 | 0.113 |
| EFNet [62] | 31.26 | 0.940 | 0.098 |

| Method | CLIPIQA↑ | MUSIQ↑ | RANKIQA↓ |
|---|---|---|---|
| *Video Reconstruction Task* | | | |
| E2VID [55] | 0.139 | 46.52 | 4.879 |
| TimeLen++ [65] | 0.144 | 48.68 | 4.325 |

**Event-based Video Reconstruction**. Finally, we assess the utility of GS2E as a benchmark for event-driven video reconstruction tasks [55, 65], including frame interpolation and intensity reconstruction. Owing to its fine-grained temporal resolution, accurate camera motion, and realistic lighting variations, GS2E provides a challenging yet structured testbed for evaluating reconstruction quality under high-speed motion. As shown in Table 3, existing models exhibit improved motion continuity and reduced ghosting artifacts when evaluated on our dataset, highlighting its effectiveness.

Table 4: RGB NVS quality before event simulation. Higher is better for PSNR/SSIM; lower is better for LPIPS.

| Scene | PSNR ↑ | | SSIM ↑ | | LPIPS ↓ | |
|---|---|---|---|---|---|---|
| | 3DGS | +Difix3D+ | 3DGS | +Difix3D+ | 3DGS | +Difix3D+ |
| Scene_1 | 29.10 | 31.64 | 0.945 | 0.953 | 0.171 | 0.162 |
| Scene_2 | 32.01 | 33.82 | 0.939 | 0.951 | 0.117 | 0.109 |
| Scene_3 | 28.98 | 30.15 | 0.936 | 0.944 | 0.143 | 0.131 |
| Scene_4 | 38.41 | 38.63 | 0.959 | 0.960 | 0.158 | 0.159 |
| Scene_5 | 35.79 | 36.29 | 0.967 | 0.969 | 0.071 | 0.064 |
| Scene_6 | 31.81 | 33.32 | 0.934 | 0.947 | 0.318 | 0.292 |
| Scene_7 | 29.26 | 32.49 | 0.942 | 0.968 | 0.151 | 0.137 |
| Scene_8 | 32.88 | 34.21 | 0.928 | 0.941 | 0.153 | 0.126 |
| Scene_9 | 30.87 | 31.95 | 0.928 | 0.940 | 0.240 | 0.228 |
| Scene_10 | 30.67 | 32.48 | 0.895 | 0.914 | 0.290 | 0.291 |
| Scene_11 | 35.29 | 36.03 | 0.955 | 0.956 | 0.261 | 0.260 |
| **Average** | **32.28** | **33.73** | **0.939** | **0.949** | **0.188** | **0.178** |

## 4.4 Ablation study

**Interplation Methods of Trajectory.** To analyze how different interpolation strategies influence the quality of synthesized event streams and their impact on downstream reconstruction tasks, we compare linear, Bézier, and cubic B-spline methods for virtual camera trajectory generation, shown as in Figure 6. While linear interpolation is efficient, its velocity discontinuities at control points can undermine temporal coherence in high-fidelity reconstruction. Cubic B-splines, by ensuring smooth higher-order continuity, yield more realistic trajectories. We thus use cubic B-spline interpolation with velocity control as the default, balancing smoothness and trajectory realism.

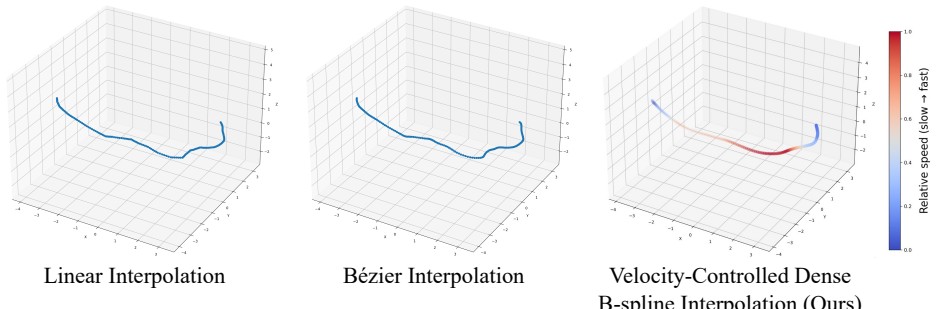

Linear Interpolation     Bézier Interpolation     Velocity-Controlled Dense
                                                   B-spline Interpolation (Ours)

Figure 6: Comparison of different interpolation methods shows that our method is smoother and has speed control capabilities.

**The Generative Refinement Module** We evaluate the effect of the RGB enhancement on static scenes from DL3DV and MVImgNet using PSNR, SSIM, and LPIPS. Table 4 reports per-scene results for 3DGS and 3DGS&Difix3D+[71]. The enhancement yields consistent gains across 11 scenes: average *PSNR* increases by **+1.45 dB** (min +0.22 dB, max +3.23 dB), *SSIM* improves by **+0.010**, and *LPIPS* decreases by **0.010** on average. These results indicate that RGB novel views entering the event simulator are of sufficiently high fidelity and reduced artifact levels, addressing the risk of artifact-induced spurious events.

## 5 Conclusion

We introduced GS2E, a large-scale dataset that synthesizes high-fidelity event streams from sparse multi-view RGB. Our pipeline couples 3DGS reconstruction with a physically grounded simulator, featuring adaptive trajectory interpolation and contrast-threshold modeling, and employs a diffusion-based RGB refinement module to reduce artifacts before event simulation. This yields temporally dense, geometry-consistent events under diverse motion and lighting. Experiments show clear gains on downstream tasks (e.g., event-based 3D reconstruction and video interpolation). Future work will incorporate exposure-aware camera models into 3DGS and extend to dynamic scenes.

## Acknowledgment

This work was supported in part by the Natural Science Foundation of China (No. 62332002, 62202014, 62425101)

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

# A Details of Velocity-Controlled Reparameterization

In our code, we provide two ways to precisely control speed. These are using continuously defined functions and a discrete speed list.

## A.1 Continuous speed function

A positive, analytic function

$$v : [0,1] \to \mathbb{R}_{>0} \quad \text{(dimensionless)},$$

sampled at normalised time $t$, directly prescribes the speed curve. In the released dataset we adopt

$$v(t) = 0.25\,\sin(t) + 1.1, \quad t \in [0,1], \tag{13}$$

## A.2 Speed list

An arbitrary-length float array $\mathbf{r} = \{r_k\}_{k=0}^{L-1}$ ($r_k > 0$) is interpreted as *multipliers* of a base frame rate $f_{\text{base}} = 2400$ fps: the $k$-th temporal segment $[t_k, t_{k+1}]$ of length $\Delta T = T/L$ is rendered at $f_k = r_k f_{\text{base}}$. To obtain a *continuous* speed curve we blend neighbouring segments with a cubic B-spline[1] in a $2\tau$-second window centred at each boundary,

$$v(t) = \text{Bspline}(t; \mathbf{r}, \tau), \qquad \tau \simeq 0.1\,\Delta T.$$

## A.3 From speed curve to arc-length samples

Let $M$ be the desired number of interpolated frames, we sample the chosen speed interface on a uniform grid $t_j = j/(M-1)$:

$$u_j = v(t_j), \quad j = 0, \ldots, M-2; \tag{14}$$

$$\Delta s_j = \frac{u_j}{\sum_{k=0}^{M-2} u_k}\, S; \tag{15}$$

$$s_0 = 0, \quad s_{j+1} = s_j + \Delta s_j. \tag{16}$$

Equation (15) rescales the sampled speeds so that $\sum_j \Delta s_j = S$, ensuring the full geometric path is covered.

## A.4 Evaluating the spline

Each interpolated pose $\tilde{P}_j = (\tilde{R}_j, \tilde{\mathbf{T}}_j)$ is obtained by querying the spline at the renormalised arc-length $s_j^\star$:

$$\tilde{P}_j = \mathcal{P}(s_j), \qquad j = 0, \ldots, M-1.$$

Because $s_{j+1} - s_j \propto v(t_j)$, the linear and angular velocities of the discrete trajectory $\{\tilde{P}_j\}$ follow the prescribed speed profile with frame-level accuracy.

## A.5 Practical remarks

- **Choice of interface.** The analytic form (13) is convenient for dataset-level consistency; the speed list form offers frame-accurate speed control for bespoke sequences.

- **Continuity.** Both interfaces yield a $C^2$ speed curve, hence the final trajectory is at least $C^1$, avoiding jerk during rendering.

- **Complexity.** The whole pipeline is linear in $N+M$ and is CPU-friendly ($< 0.5\,\mu$s per interpolated pose).

**Summary.** Either a compact analytic law (13) or an arbitrary-length speed list can be mapped, via Eq. (15), to B-spline arc-length samples, providing reliable and precise control over camera velocity for every rendered frame.

---

[1]Order 3 suffices to reach $C^2$ continuity while keeping local support.

## B   Details of choosing the contrast threshold

In our experiments, we found that when we set the contrast threshold $c \leq 0.75$, visible floater artifacts appeared during the visualization of the event stream. These artifacts occur when the viewpoint changes and certain Gaussians—originally situated in the background and expected to be occluded—are mistakenly treated as part of the visible foreground. This misclassification leads to variations in illumination that induce apparent voltage changes, which the simulator erroneously interprets as valid event triggers. As a result, the synthesized event stream contains non-physical textures, manifesting as spurious structures or noise in the visualization. As shown in the figure 7, once we raise $c$ to 1 or higher, the floater becomes almost invisible.

It is worth noting that when the contrast threshold is set too low, according to the research results in [51], it will lead to a loss of dynamic range. Therefore, in this paper, we tend to set a larger $c$ to solve both problems simultaneously. To ensure that events are not overly sparse and sufficient information integrity is retained, the GS2E dataset was simulated with the parameter setting $c = 1$.

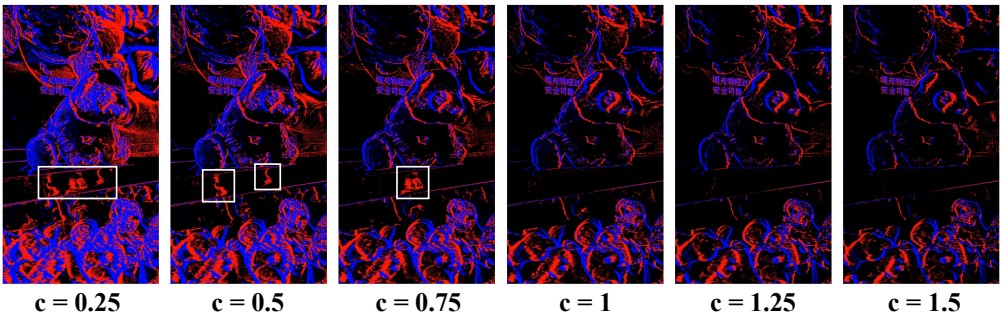

| c = 0.25 | c = 0.5 | c = 0.75 | c = 1 | c = 1.25 | c = 1.5 |

Figure 7: Selecting the same viewpoint and time window(1000 us), visualize events simulated from 3DGS with different contrast threshold($c$) values. The results show that when $c \leq 0.75$, error events generated by floater Gaussians can be seen on the integral event diagram, while this phenomenon is greatly alleviated when $c \geq 1$.

## C   Implementation Details

For the reconstruction stage, we carefully collect approximately 2k high-quality multi-view images from 2 public datasets: MVImageNet [80] and DL3DV [38]: we choose and render 1.8k scenes from MVImageNet and 100 scenes from DL3DV. We employ the official interplementation version of 3DGS [31] in the original setting. For the event generation stage, we utilize the DVS-Voltmeter simulator [37] to synthesize events from the rendered RGB sequences. We adopt the following sensor-specific parameters to closely mimic the behavior of real DVS sensors: the ON and OFF contrast thresholds are both set to $\Theta_{\text{ON}} = \Theta_{\text{OFF}} = 1$ as default. The dvs camera parameters are calibrated as $k_1 = 0.5, k_2 = 1e-3, k_3 = 0.1, k_4 = 0.01, k_5 = 0.1, k_6 = 1e-5$, following the original DVS-Voltmeter setting. These control the brightness-dependent drift $\mu$ and variance $\sigma^2$ of the stochastic process, which determine the polarity distribution and the inverse-Gaussian timestamp sampling for each event.

All events are simulated at 2400 FPS temporal resolution and stored with microsecond timestamps for high-fidelity spatio-temporal alignment. The overall process are conducted on a workstation equipped with $8 \times$NVIDIA RTX 3090 GPUs. The selected MVImageNet clip images vary in size, but most are approximately 1080p in resolution. When training 3DGS on MVImageNet, each scene takes an average of 16 minutes. For the camera pose upsampling and trajectory control stage, using an interpolation factor of $\gamma = 5$, the strategy `ada_speed`, and the velocity function $v(t) = 0.25 \sin(t) + 1.1$, the average runtime per scene is approximately 45 seconds.

During event simulation, we adopt the same camera parameter configuration as mentioned previously. However, the simulation time varies significantly depending on the motion amplitude and speed of the camera, as well as the scene complexity, making it difficult to estimate a consistent runtime.

For the DL3DV dataset, each scene contains 300–400 images. To ensure higher reconstruction and rendering quality, as well as to generate longer event streams, we do not downsample the input image

resolution, nor do we slice the image or event sequences. Using the same hardware configuration as with MVImageNet, the average per-scene training time is approximately 27 minutes, and the rendering time is around 41 minutes.

## D  Existing Event-based 3D Reconstruction Datasets

To contextualize the contribution of GS2E, Table 5 provides a comprehensive comparison of existing event-based 3D datasets and 3D reconstruction methods [43, 9, 52, 2, 82, 63, 87, 53, 58, 79, 89, 73, 77, 24, 18, 57, 72, 1, 78, 81, 41, 42, 86, 68, 25, 88, 39, 15, 35, 49, 4, 22, 66, 17, 11, 85, 44, 21]. We categorize these into **static scenes** and **dynamic scenes**, based on whether the underlying geometry remains constant or involves temporal variation.

**Attributes.**    Each dataset is evaluated along key axes:

- **Data Type:** Whether sharp and/or blurry RGB frames are provided. Blurry frames support deblurring tasks, while sharp ones aid in geometry fidelity.

- **Scene Num / Scale:** Number of distinct scenes and their spatial scope (object-level vs. medium/large indoor scenes).

- **GT Poses:** Availability of ground-truth camera extrinsics.

- **Speed Profile:** Whether camera motion follows uniform or non-uniform velocity.

- **Multi-Trajectory:** Whether each scene supports multiple trajectory simulations, enabling consistent multi-view observations.

- **Device:** Capture source—real event sensors (e.g., DAVIS346C, DVXplore) or simulated streams (e.g., ESIM, Vid2E, V2E).

- **Data Source:** Origin of the base scene data (e.g., NeRF renderings, Blender, Unreal Engine, or real-world scenes).

**Key Findings.**    We observe that existing datasets are limited in several aspects:

- Most datasets focus on small-scale, object-centric scenes with limited spatial or temporal diversity.

- Simulators typically use simplified trajectories and fixed contrast thresholds, which constrain realism.

- Real event data remains scarce and often lacks consistent trajectory coverage or paired ground truth.

- Multi-trajectory support is rare, impeding evaluation under view-consistency and generalization settings.

**Positioning of GS2E.**    Our proposed GS2E benchmark is designed to address these limitations by:

- Leveraging 3D Gaussian Splatting to reconstruct photorealistic static scenes from sparse real-world RGB inputs.

- Generating controllable, dense virtual trajectories with adaptive speed profiles and multiple interpolated paths per scene.

- Synthesizing events via a physically-informed simulator that incorporates realistic contrast threshold modeling.

- Supporting both object- and scene-level scales with consistent multi-view alignment and temporal density.

By filling the gaps in scale, realism, and trajectory diversity, GS2E enables more robust evaluation of event-based 3D reconstruction and rendering methods.

# E  Limitation and Broader impacts

**Limitation.** While GS2E provides high-fidelity, geometry-consistent event data under a wide range of camera trajectories and motion patterns, it remains fundamentally limited by its reliance on rendered RGB images from 3DGS. Specifically, the current pipeline inherits the photometric constraints of 3D Gaussian Splatting , which may not faithfully replicate extreme illumination conditions such as overexposure or underexposure. As a result, scenes with very low light or high dynamic range may not be accurately modeled in terms of event triggering behavior. Additionally, our framework currently assumes static scenes; dynamic object motion is not yet modeled. In future work, we plan to extend the simulator by incorporating physically-realistic camera models into the 3DGS rendering pipeline, enabling explicit control over exposure, tone mapping, and sensor response curves to better approximate real-world lighting variability.

**Broader impacts.** This work introduces a scalable, geometry-consistent synthetic dataset for event-based vision research. On the positive side, it lowers the barrier for training high-performance models in domains such as autonomous driving, robotics, and augmented reality, where event-based sensing offers advantages under fast motion or challenging lighting. By providing a flexible, physically-grounded simulation framework, the work supports reproducible and ethical AI development. On the negative side, improved realism in synthetic event data may inadvertently enable misuse such as generating adversarial inputs or synthetic surveillance data. These risks are mitigated by the dataset's academic licensing and transparency in its construction pipeline. Furthermore, the data generation framework may raise privacy concerns if adapted for real-scene reproduction, which warrants further community discussion and the adoption of usage safeguards.

# F  License of the used assets

- **3D Gaussian Splatting [31]**: A publicly available method with its dataset released under the CC BY MIT license.
- **MVImgNet [80]**: A publicly available dataset released under the CC BY 4.0 license.
- **DL3DV [38]**: A publicly available dataset released under the CC BY 4.0 license.
- **GS2E**: A publicly available dataset released under the CC BY MIT license.

| Method | Column | Type | Color Frame Blurry | Color Frame Sharp | Scene Num | Scene Scale | GT poses | Speed | Multi-Trajectory | Device | Data Source |
|---|---|---|---|---|---|---|---|---|---|---|---|
| **Static Scenes** | | | | | | | | | | | |
| Event-NeRF [57] | CVPR 2023 | Synthetic | × | ✓ | 7 | object | ✓ | Uniform | × | Blender+ESIM | NeRF |
| E2NeRF [52] | ICCV 2023 | Synthetic | ✓ | ✓ | 7 | object | ✓ | Uniform | × | Blender+ESIM | NeRF |
| Robust e-NeRF [41] | ICCV 2023 | Real | ✓ | × | 5 | medium&large | ✓ | Uniform | × | DAVIS346C | Author Collection |
| Deblur e-NeRF [42] | ECCV 2024 | Synthetic | × | ✓ | 7 | object | ✓ | Non-Uniform | ✓ | Blender+ESIM | NeRF |
| EvaGaussian [79] | Arxiv 2024 | Synthetic | ✓ | × | 9 | medium&large | ✓ | Uniform | ✓ | Blender+ESIM | NeRF, Deblur-NeRF+Author Collection |
| PAEv3D [69] | ICRA 2024 | Real | ✓ | × | 5 | medium&large | × | Uniform | × | DAVIS346C | Author Collection |
| EvDeblurRF | CVPR 2024 | Synthetic | ✓ | ✓ | 6 | object | × | Uniform | × | DVXplore event camera | Author Collection |
| EvGGS [68] | ICML 2024 | Synthetic | ✓ | ✓ | 4 | medium | ✓ | Uniform | × | Blender+ESIM | Deblur-NeRF |
| IncEventGS [24] | CVPR 2025 | Synthetic | × | ✓ | 5 | medium | ✓ | Uniform | × | DAVIS346C | Author Collection |
| E-3DGS [82] | 3DV 2025 | Synthetic | × | ✓ | 64 | object | ✓ | Uniform | × | Blender+V2E | Author Collection |
| AE-NeRF [9] | AAAI 2025 | Synthetic | × | ✓ | 6 | large | ✓ | Non-Uniform | × | Vid2E | Replica |
| EF-3DGS [36] | Arxiv 2024 | Synthetic | × | ✓ | 3 | medium | ✓ | Non-Uniform | × | - | UnrealEgo |
| EF-3DGS [36] | Arxiv 2024 | Synthetic | × | ✓ | 3 | medium | ✓ | Non-Uniform | × | - | UnrealEgo |
| LSE-NeRF [63] | Arxiv 2024 | Synthetic | ✓ | ✓ | 9 | large | ✓ | Uniform | × | Vid2E | Tanks and Temples |
| LSE-NeRF [63] | Arxiv 2024 | Real | ✓ | ✓ | 10 | medium&large | ✓ | Non-Uniform | - | Prophesee EVK-3 HD + Blackfly S (GigE) | Author Collection |
| **GS2E** | **Submission** | **Synthetic** | ✓ | ✓ | **1150** | **medium&large** | ✓ | **Non-Uniform** | ✓ | **3DGS + DVS-Voltmetre** | **Author Collection** |
| **Dynamic Scenes** | | | | | | | | | | | |
| DE-NeRF [43] | ICCV 2023 | Synthetic | × | ✓ | 3 | object | ✓ | Uniform | ✓ | Blender+ESIM | Author Collection |
| EvDNeRF [1] | CVPR 2024 Workshop | Real | × | × | 6 | medium&large | ✓ | Uniform | × | Samsung DVS Gen3, DAVIS 346C | Color Event Camera, HS-ERGB(Timeslen) |
| Dynamic EventNeRF [58] | CVPR 2025 Workshop | Synthetic | × | ✓ | 3 | object | ✓ | Uniform | ✓ | Blender+ESIM | Kubric |
| Dynamic EventNeRF [58] | CVPR 2025 Workshop | Real | × | ✓ | 16 | medium&large | ✓ | Uniform | ✓ | DAVIS346C | Author Collection |

Table 5: Comparison of existing event-based 3D reconstruction datasets, categorized by scene type, motion profile, sensor modality, and simulation pipeline.

