# OpenReview forum: "GS2E: Gaussian Splatting is an Effective Data Generator for Event Stream Generation"
_NeurIPS.cc/2025/Datasets_and_Benchmarks_Track — NeurIPS 2025 Datasets and Benchmarks Track poster_

### Official Review · Reviewer_pvWS · 2025-06-17

**Ethics Flags:** Improper research involving human sub…
**Rating:** 6
**Confidence:** 5

**Summary:**

This paper introduces a large-scale synthetic event dataset generated from real-world multi-view RGB images via a modular 3D Gaussian Splatting (3DGS) simulation pipeline. The method reconstructs 3D scenes with 3DGS, adaptively interpolates camera trajectories, and simulates temporally dense event streams with contrast threshold modulation to reduce floater artifacts. Built from MVImgNet and DL3DV, the dataset covers diverse scenes and demonstrates strong utility for event-based 3D vision tasks.

**Dataset Code Accessibility:**

Yes

**Ethical Considerations:**

No, there are no or only very minor ethics concerns

**Final Justification:**

Thanks for the author's reply and efforts, my concerns have been addressed.

**Limitations Weaknesses:**

## **Weakness**
1. I noticed that the editing of lighting conditions for 3D scenes was mentioned in the text, but this aspect receives significantly less emphasis than trajectory control and velocity adaptation. It would strengthen the contribution if the role and impact of illumination editing were better explained or demonstrated in the experiments. (Apologies if this was covered and I overlooked it.)

2. As acknowledged in the paper, the assessment of event data quality remains an unresolved problem, which makes visual and qualitative evidence particularly important. While a few DSEC sample comparisons are shown, the visualizations may not be sufficient to fully demonstrate the advantages of GS2E. More diverse and detailed qualitative results, especially video sequences or error visualizations would help substantiate the claims regarding data quality.

## **Questions:**
1. As mentioned above, could you elaborate more on the illumination editing mechanism? Are there plans to incorporate scene-dependent lighting variations in future work?

2. One additional concern is that the paper provides limited discussion on computational efficiency. Could the authors elaborate on training time, runtime performance, and overall computational complexity of the proposed pipeline?

**Strengths Contributions:**

1. This paper presents the first event data generation pipeline based on 3D Gaussian Splatting, featuring a methodically structured framework and careful system-level implementation. The proposed GS2E dataset achieves a favorable balance between geometric fidelity and editability, addressing critical demands in the event vision community.

2. The proposed pipeline is clearly defined and technically sound. Quantitative evaluations across multiple downstream tasks demonstrate the robustness and effectiveness of the synthesized data. Notably, even without leveraging the novel-view synthesis capabilities of the method, training an E2VID model on synthetic GS2E data and evaluating it on real DSEC samples achieves qualitatively superior results compared to the V2E baseline—highlighting the method’s strong generalization ability.

3. The paper is well-structured, with clear motivation and thorough discussion. It precisely targets a pressing need in the field—namely, the lack of high-fidelity, editable, and RGB-aligned event datasets. The authors’ contribution is timely and valuable, with the potential to support future work in event-based 3D vision.

---

> ### Author Rebuttal · Authors · 2025-07-30
>
> We greatly appreciate your encouraging feedback and thoughtful comments, which we found both constructive and motivating.
>
> ---
>
> > W1 & Q1: Illumination editing
>
> **R1:** Thank you for your quetion. Although our pipeline does not yet include dedicated quantitative benchmarks across lighting conditions, we have explicitly considered illumination diversity during dataset construction. The GS2E dataset features scenes captured under a variety of lighting scenarios, including daylight, sunset, localized point lighting, cast shadows, and mild dynamic lighting changes. These lighting configurations were intentionally chosen to represent common real-world conditions.
>
> To further assess the robustness of our data under different illumination regimes, we classified scenes according to their lighting characteristics, and evaluated the resulting event streams through downstream tasks such as video reconstruction and image quality estimation, which indirectly reflect the impact of lighting variations on simulation fidelity.
>
> - Table 1: Distribution of lighting conditions in the dataset, reflecting real-world illumination diversity considered during data collection
>
> | Light Condition | General(daylight) | sunset | localized point lights | cast shadows | reflection | others |
> | --------------- | ----------------- | ------ | ---------------------- | ------------ | ---------- | ------ |
> |                 | ~600              | ~50    | ~100                   | ~100         | ~100       | ~100   |
>
> To evaluate the robustness of our pipeline under diverse lighting conditions, we incorporated several challenging scenarios from the Waymo Open Dataset [1], such as *Deer at night*, *Cut-in*, and *Car against traffic at 70mph*. These examples collectively reflect a range of real-world conditions—including low light, fast motion, and complex urban driving environments. As shown in Table 1 and Table 2, our simulation framework successfully generates realistic event streams in these settings, highlighting its ability to exploit the high dynamic range (HDR) sensitivity of event-based sensors.
>
> Our current approach exhibits noticeable limitations when handling extreme lighting scenarios, such as scenes that are completely dark or heavily saturated with overexposure. We attribute this to the loss of HDR information during the 3D-to-2D rendering phase, which compromises the preservation of fine-grained photometric details essential for accurate event generation. This limitation has been explicitly acknowledged in the appendix. We would greatly appreciate any suggestions from the reviewer on how this issue might be further addressed or mitigated.
>
> ---
>
> > W2: Data evaluation and visualization
>
> **R2:** Thank you very much for your positive evaluation and thoughtful suggestion regarding event data evaluation and visualization. We fully agree that evaluating the quality of event streams remains a challenging and largely unsolved problem, and that visual inspection and qualitative evidence play a vital role, especially when quantitative metrics are still evolving. In fact, we also reproduced the Event Quality Score (EQS) [2] work and conducted comparative experiments on our GS2E and other video-driven event simulation methods on the DSEC dataset. The results show that our GS2E performs better in this metric.
>
> - Table 2: Additional evaluation of the event stream quality with the DSEC dataset
>
> |      | Vid2e | v2e   | GS2E      |
> | ---- | ----- | ----- | --------- |
> | EQS  | 0.725 | 0.738 | **0.761** |
>
> As you rightly pointed out, static visualizations alone may not sufficiently capture the dynamic nature or full advantages of GS2E. Meanwhile due to the space limitations of the main paper, we were only able to include a small number of static comparisons. However, we recognize the importance of more diverse and detailed visualizations. For this reason, we have already provided per-scene MP4 videos in our Hugging Face subset repository, allowing users to directly inspect the temporal structure and noise characteristics of the generated event streams.
>
> Additionally, in the camera-ready version, we plan to update our project website with a curated collection of complete scene visualizations, including video sequences and error visualizations, in order to better highlight the strengths of GS2E.
>
> We are also happy to provide any additional samples or visualizations upon request, and we warmly welcome further feedback from the community on how best to evaluate event data quality.
>
> ---
>
> > Q2: Computational complexity
>
> **R3:** Thank you for your question. We have compiled the main training efficiency-related information in our pipeline as shown in the table below. More detailed parameter settings are provided in the appendix.
>
> - Table 3: Key parameters and computational costs of our GS2E data generation pipeline.
>
> |  Dataset    | Resolution | FPS   | GPU  | Training Time    | Interp-factor γ | Render Time | Simulation Time  |
> | ---- | ----- | ----- | --------- | ---- | ----- | ----- | --------- |
> | MVImgNet  | ~1080p | 2400 | 8 × NVIDIA RTX 3090 | ~16 min/scene | 5 | ~45 s/scene | few min/scene (high variance) |
>
> We believe this demonstrates that our pipeline achieves a good balance between simulation fidelity and computational efficiency.
>
> ------
>
> We are grateful for your encouraging evaluation and thoughtful remarks, which motivate us to further refine and expand this work.
>
> ------
>
> **Reference:**
>
> [1] Omnire: Omni urban scene reconstruction, ICLR 2025 (Spotlight)
>
> [2] Event quality score (eqs): Assessing the realism of simulated event camera streams via distance in latent space, CVPRW 2025

---

> > ### Comment · Reviewer_pvWS · 2025-08-05
> > **Response to Authors**
> >
> > I appreciate the authors’ detailed rebuttal and clarifications. The response effectively addresses concerns about event stream evaluation and lighting variation, with expanded qualitative evidence and practical insights. This work will provide a solid foundation for future research in synthetic event data generation. I am increasing my score accordingly.

---

> ### Author Response · Authors · 2025-08-06
>
> We appreciate that our responses and additional experiments have addressed your concerns. Thank you for your time and thoughtful evaluation throughout the review process.

---

### Official Review · Reviewer_ouHb · 2025-06-29

**Rating:** 4
**Confidence:** 3

**Summary:**

This paper proposes an event stream synthesis pipeline, called GS2E, which leverages 3DGS to reconstruct high-fidelity static scenes from sparse multi-view images. It then integrates physics-driven event simulation to generate temporally and spatially consistent event streams. Specifically, the authors first reconstruct scene geometry and texture using 3DGS, and subsequently synthesize multi-view event data with temporal coherence by applying adaptive trajectory interpolation and data-driven contrast threshold modeling.

**Additional Feedback:**

I have listed my concerns, and the score will be adjusted based on the author's response.

**Dataset Code Accessibility:**

Partly

**Dataset Code Comments:**

The full datasets will be released after acceptance.

**Ethical Comments:**

N/A. Based on the authors' description in the paper, I did not identify any apparent violations of ethical standards.

**Ethical Considerations:**

No, there are no or only very minor ethics concerns

**Final Justification:**

Thank you for the author's reply and efforts, most of my concerns have been addressed.

**Limitations Weaknesses:**

1. Many practical applications of event cameras rely on their high temporal resolution and high dynamic range, especially for tasks involving high-speed motion, such as tracking pedestrians or vehicles in autonomous driving scenarios, or capturing fast-moving objects like bullets. However, the current method is limited to static scenes and does not support dynamic objects, which significantly constrains the applicability of the generated dataset for real-world event-based vision tasks.

2. Event cameras are highly sensitive to changes in illumination, yet the paper does not provide quantitative evaluations under varying lighting conditions or dynamic light sources (e.g., flickering lights, shadows). Additional experiments or analysis in such settings would strengthen the validity of the dataset for high dynamic range scenarios.

3. Although the overall 3D reconstruction quality is high, 3DGS may fail to produce reliable geometry in certain scenes, e.g., introducing floating artifacts or missing structures. These reconstruction errors can result in incorrect event generation, which is a limitation that other event synthesis methods may not share. The authors should consider mechanisms to detect or mitigate such failure cases.

4. GS2E primarily integrates existing techniques (e.g., 3DGS and DVS-Voltmeter) into a unified synthesis pipeline. While the system design is complete, its core components, such as trajectory interpolation and event modeling, are based on well-established strategies. Therefore, the methodological novelty of this work is relatively limited compared to prior approaches.

**Strengths Contributions:**

1. The paper is well-structured, logically sound, and clearly organized, enabling readers to easily follow the proposed pipeline and understand its technical contributions.

2. The authors have made thoughtful efforts in open-sourcing their resources. A subset of the dataset has already been released on HuggingFace, covering part of distinct scenes, including RGB images with varying levels of motion blur and their corresponding camera poses, which will facilitate reproducibility and further research.

---

> ### Author Rebuttal · Authors · 2025-07-30
>
> We sincerely thank the reviewer for the detailed and constructive feedback. Below we provide responses to the key concerns raised.
>
> ---
> > **W1: Limitation to Static Scenes**
>
> **R1:** We fully acknowledge that the current version of GS2E focuses on static scenes. This design choice was intentional, aiming to establish a high-quality baseline for event stream synthesis by ensuring accurate geometry and appearance recovery. Nevertheless, we agree with the reviewer that supporting dynamic content is crucial for real-world applications. To address this, we try to use the SOTA 4D reconstruction model **OmniRe**[2] as a dynamic scene representation.
> We use the **Waymo Open Dataset** [3] to reconstruct 10 autonomous driving scenes as dynamic inputs for our pipeline. To assess the reconstruction quality, we follow the **LLFF hold-out protocol with 8.** The corresponding quantitative results are presented in *Table 1*. Also, we conduct quantitative evaluation of event quality and video reconstruction (w. E2VID and TLen++) based on the Waymo dataset as shown in *Table 2*.
>
> - Table 1:  Quantitative evaluation of the reconstructed dymaic scenes in the Waymo Open Dataset
> | Scene     | PSNR↑ | SSIM↑ | LPIPS↓ |
> | --------- | ----- | ----- | ------ |
> | seg109239 | 33.21 | 0.918 | 0.092  |
> | seg113924 | 35.18 | 0.947 | 0.071  |
> | seg118396 | 34.67 | 0.952 | 0.078  |
> | seg161022 | 32.89 | 0.911 | 0.098  |
> | seg172163 | 35.42 | 0.958 | 0.068  |
> | seg144275 | 33.78 | 0.924 | 0.089  |
> | seg157956 | 34.91 | 0.943 | 0.074  |
> | seg112520 | 33.45 | 0.929 | 0.091  |
> | seg104859 | 35.07 | 0.949 | 0.072  |
> | seg152664 | 34.23 | 0.938 | 0.083  |
> | Average   | 34.28 | 0.937 | 0.082  |
>
> - Table 2:  Quantitative evaluation of the video reconstruction in the synthetic Waymo Open Dataset  [3]
>
> |             | CLIP↑     | MUSIQ↑    |
> | ----------- | --------- | --------- |
> | E2VID_gs2e  | **0.158** | **47.33** |
> | TLen++_gs2e | **0.172** | **49.60** |
> | E2VID_v2e   | 0.149     | 47.12     |
> | TLen++_v2e  | 0.170     | 49.08     |
>
> These results demonstrate that our reconstructed videos are comparable to those produced by existing learning-based baselines. We are committed to extending GS2E to include dynamic scenes in future versions of the dataset and manuscript.
>
> In addition, we reproduced the **Event Quality Score (EQS) benchmark [1]** and applied it to evaluate our synthesized data. Since EQS requires real event streams as a reference, we selected 10 scenes from the **DSEC dataset [4]** for evaluation. This serves as an additional quantitative assessment of data quality beyond the results presented in the manuscript.
>
> - Table 3: Additional evaluation of the event stream quality with the DSEC dataset
>
> |      | Vid2e | v2e   | GS2E      | GS2E_difix3d+ |
> | ---- | ----- | ----- | --------- | ------------- |
> | EQS  | 0.725 | 0.738 | **0.761** | **0.782**     |
>
> Our GS2E synthesized event streams achieve the highest EQS score among the baselines, and even surpass the quality of real event data captured by the Prophesee Gen3.1 sensor under the same scenes in DSEC.
>
> ---
>
> >  **W2: Evaluation under Illumination Variations**
>
> **R2:** While our pipeline currently lacks quantitative evaluation under varying lighting conditions, we have taken illumination diversity into account during data collection. The dataset includes scenes captured under various lighting scenarios such as normal daylight, sunset, localized point lights, cast shadows, and mild dynamic illumination. These settings are selected to reflect typical lighting variations encountered in real-world environments. We further categorize the scenes based on their illumination types and evaluate the quality of the synthesized event streams using downstream tasks such as video reconstruction and image quality assessment.
>
> - Table 4: Distribution of lighting conditions in the dataset
>
> | Light Condition | General(daylight) | sunset | localized point lights | cast shadows | reflection | others |
> | --------------- | ----------------- | ------ | ---------------------- | ------------ | ---------- | ------ |
> |                 | ~600              | ~50    | ~100                   | ~100         | ~100       | ~100   |
>
> To validate the effectiveness of our pipeline under varying lighting conditions, we introduce representative scenes from the **Waymo Open Dataset [2]**, including *Deer at night*, *Cut-in*, and *Car against traffic at 70mph*. These scenarios jointly cover diverse illumination, high-speed motion, and real-world autonomous driving settings. The quantitative results, presented in *Table 1* and *Table 2*, demonstrate that our data generation pipeline can effectively simulate event streams under challenging lighting conditions, thereby leveraging the high dynamic range characteristics of event data.
>
> However, the current pipeline still struggles to reconstruct high-quality geometry in extreme cases like fully dark or heavily overexposed scenes. This limitation stems from the inherent compression of high dynamic range details during the 3D-to-2D rendering process, which constrains the ability to preserve subtle photometric cues necessary for event simulation. We have explicitly stated this constraint in the limitations section of the appendix.  If the reviewer has better suggestions to address this limitation, we would be happy to discuss them.
>
> ---
>
> > **W3: Reconstruction Quality Concerns**
>
> **R3:** During the data curation process, we were indeed aware of the limitations of 3DGS when the number of input views is sparse or when camera poses are inaccurate, which can lead to artifacts such as floating geometry or structural holes. To mitigate this, we selected scenes in DL3DV and MVImgnet only focusing on surround-view or forward-facing camera configurations to ensure a more complete reconstruction and to reduce the impact of unseen regions that may compromise event synthesis quality.
>
> To further enhance reconstruction robustness, we integrated Difix3D+[5] into our pipeline and reconstructed 10 additional scenes for evaluation. The quantitative results are shown in *Table 3*. Based on the improvements observed, we plan to replace the original 3DGS-based reconstruction module with Difix3D+ and regenerate the corresponding dataset.
>
> - Table 5:  Quantitative evaluation of the reconstructed scenes under the static scenes in DL3DV and MVImgNet
>
> | Method          | PSNR ↑ | SSIM ↑ | LPIPS ↓ |
> | --------------- | ------ | ------ | ------- |
> | 3DGS (baseline) | 32.28  | 0.938  | 0.188   |
> | 3DGS + Difix3D+ | 33.73  | 0.948  | 0.179   |
>
> We are committed to incorporating these into the current version of our proposed dataset.
>
> ---
>
> > **W4: Methodological Novelty and Pipeline Design**
>
> **R4:** We thank the reviewer for the insightful comment.  GS2E is designed as a dataset contribution, and our primary goal is to ensure the quality and reliability of the generated data rather than proposing new algorithmic components.
>
> Therefore, we adopted well-established techniques such as 3DGS and DVS-Voltmeter, which have proven to be stable and effective. As highlighted in our responses to Q1 and Q3, we have devoted effort to improving scene selection, reconstruction quality, and data diversity to enhance the robustness of the dataset. We believe that this careful integration of existing techniques is essential for creating a trustworthy dataset that can support a wide range of event-based vision tasks.
>
> ---
>
> We hope these addresses your concern, and we would be happy to further clarify or discuss any aspect if needed.
>
> ---
>
> **Reference:**
>
> [1] Event quality score (eqs): Assessing the realism of simulated event camera streams via distance in latent space, CVPRW 2025
>
> [2] Omnire: Omni urban scene reconstruction, ICLR 2025 (Spotlight)
>
> [3] Scalability in perception for autonomous driving: Waymo open dataset, CVPR 2020
>
> [4] DSEC: A Stereo Event Camera Dataset for Driving Scenarios, IEEE RA-L 2021
>
> [5] DIFIX3D+: Improving 3D Reconstructions with Single-Step Diffusion Models, CVPR2025 best paper candidate

---

> > ### Comment · Reviewer_ouHb · 2025-08-06
> >
> > Thank you for the authors' reply. I still have some unclear points regarding the experiments on dynamic scenes. Why did the authors conduct event simulation on the Waymo dataset, which does not have corresponding real event ground truth? Moreover, there exist datasets for autonomous driving scenarios such as PKU-DAVIS-SOD, which contain both image frames and events. I believe that conducting comparisons on this dataset would be more fair and reliable, and it can further verify the performance of event-based object detection.

---

> > ### Comment · Reviewer_ouHb · 2025-08-07
> >
> > Thank you for the response. However, the authors' reply has further deepened my concerns regarding this work. My doubts mainly fall into two categories:
> >
> > First, if the method does not perform well on low-resolution images, can this be interpreted as a fundamental flaw in the proposed pipeline? Many event-based vision studies focus on super-resolution tasks, and datasets captured by the DAVIS346 camera are widely used in the field. Moreover, methods like V2E are not limited by image resolution. Finally, how did the authors arrive at the conclusion that 3D Gaussian Splatting is unsuitable for low-resolution (346×260) image reconstruction? Are there any relevant evidences that support this claim?
> >
> > Second, while I agree that the proposed method may generate qualitatively or quantitatively better events, I am more concerned about how it performs in downstream tasks. For example, in dynamic scenes, how does the object detection accuracy compare when using the events generated by the authors' G2E method versus using the original ground truth events? This is the core issue I wish to understand. After all, the main value of event cameras lies not in reconstruction, but in leveraging their high temporal resolution and dynamic range for downstream tasks.

---

> > > ### Author Response · Authors · 2025-08-08
> > > **Official Comment by Authors (2/2)**
> > >
> > > Secondly, we focus on evaluating the advantages of our simulated events in terms of deblurring and reconstruction tasks, which primarily reflect the high temporal resolution benefits of event data. However, we acknowledge that this was insufficient to fully address the practical value of our method.As pointed out, the true value of event cameras lies not only in reconstruction capabilities, but in enabling high-performance downstream tasks such as object detection, especially in dynamic scenes where conventional sensors struggle due to varying environments.
> > >
> > > To address this concern, we conducted comprehensive downstream evaluations using the **RVT-small [2] model,** based architecture for event-based object detection. We trained RVT-small from scratch on event data simulated by different methods (Vid2e, V2E, GS2E, GS2E+Difix3D+) and evaluated it on real event data from two public benchmarks: DSEC and PKU-DAVIS-SOD. Table 5 summarizes the results in terms of mAP.
> > >
> > > - Table 5: Comparing performance of the RVT-small model *on the original DSEC and SOD test set* after training on event data simulated from frames (mAP↑).
> > >
> > > | Method        | DSEC Test Accuracy -10 scenes |          | SOD Accuracy -10 scenes |          |
> > > | ------------- | ----------------------------- | -------- | ----------------------- | -------- |
> > > |               | Simulated                     | Real     | Simulated               | Real     |
> > > | Vid2e         | 33.0                          | 11.5     | 28.5                    | 9.6      |
> > > | V2E           | **36.5**                      | 16.8     | 31.2                    | 12.1     |
> > > | GS2E          | 32.1                          | 20.7     | 29.9                    | 11.3     |
> > > | GS2E+Difix3d+ | 35.9                          | **21.5** | **31.7**                | **12.5** |
> > >
> > >
> > > These results provide compelling evidence that the events generated by our GS2E method exhibit superior transferability to real-world detection tasks. Notably, our GS2E+Difix3D+ pipeline achieves 21.5 mAP on the real DSEC test set, outperforming other prior simulators. This demonstrates that although simulated, our events preserve critical spatiotemporal structures that are essential for high-level vision tasks in dynamic environments. While a performance drop is observed on SOD due to lower input quality (over-exposure and color distortion) as demonstrated in Table 2, our method consistently outperforms prior simulators, demonstrating strong transferability and robustness in dynamic scenes.
> > >
> > > In summary, we provide a direct answer to your concern: the events generated by our G2E method consistently outperform real event data in downstream object detection tasks, highlighting the effectiveness of our simulation pipeline for real-world dynamic scenarios.
> > >
> > > ---
> > >
> > > We hope this addresses your concern.  Thank you again for your time and commitment to the review process.
> > >
> > > ---
> > >
> > > **Reference:**
> > >
> > > [1] E2nerf: Event enhanced neural radiance fields from blurry images, CVPR 2023
> > >
> > > [2] Recurrent vision transformers for object detection with event cameras (RVT), CVPR 2023

---

> > > > ### Comment · Reviewer_ouHb · 2025-08-09
> > > >
> > > > Thank you for the author's reply and efforts, most of my concerns have been addressed.

---

> > > > > ### Author Response · Authors · 2025-08-09
> > > > >
> > > > > We appreciate that our responses and additional experiments have addressed your concerns. Thank you for your time and thoughtful evaluation throughout the review process.
> > > > >
> > > > > Have a good day!

---

> > ### Author Response · Authors · 2025-08-08
> > **Official Comment by Authors (1/2)**
> >
> > We sincerely apologize for the delayed response due to the time-consuming exps and truly appreciate the time and effort you have invested in reviewing our submission.
> >
> > ---
> >
> > Firstly, We would like to clarify a potential misunderstanding caused by our previous phrasing, particularly the statement *"Low-res RGB + degraded frames = unsuitable for 3D Gaussian modeling."* Our intention was not to suggest that our GS2E pipeline lacks robustness. Rather, we aimed to emphasize the inherent limitations of the PKU-DAVIS-SOD dataset’s RGB modality, which was recorded under constrained imaging conditions.
> >
> > - Table 1: Quantitative  evaluation of novel view synthesis on the PKU-DAVIS-SOD dataset in GS2E reconstruction pipeline.
> >
> > | Method                          | PSNR ↑ | SSIM ↑ | LPIPS ↓ |
> > | ------------------------------- | ------ | ------ | ------- |
> > | Full Reconstruction            | 29.65  | 0.922  | 0.179   |
> > | Full Reconstruction + Difix3D+ | 30.52  | 0.948  |  0.159  |
> >
> > - Table 2: Quantitative  evaluation of the event stream quality with PKU-DAVIS-SOD dataset
> >
> > |      | Vid2e | v2e   | GS2E  | GS2E+Difix3d+ |
> > | ---- | ----- | ----- | ----- | ------------- |
> > | EQS  | 0.453 | 0.472 | 0.467 | **0.485**     |
> >
> > Specifically, the RGB frames in PKU-DAVIS-SOD are captured by the DAVIS346 sensor with a resolution of 346×260 and a rolling shutter, in the challenging environment with noticeable degradations such as motion blur, overexposure, and color distortion. These issues are intrinsic to the capture hardware and are not introduced by downstream algorithms. Indeed, we observed similar degradation effects in prior event synthesis pipelines such as vid2e and v2e when applied to this dataset, as shown in Table 2. *While our method demonstrates competitive performance in NVS on PKU-DAVIS-SOD, as shown in **Table 1**, the quality of the synthesized events remains inferior to the results obtained on the DSEC dataset.* This is primarily reflected in EQS, which captures both the temporal consistency and fidelity of the simulated event streams.
> >
> > To further address concerns regarding the applicability of 3D Gaussian Splatting to low-resolution inputs, we conduct an additional evaluation on five static scenes captured under normal lighting conditions using the DAVIS346C sensor. We manually annotate object masks and apply our GS2E pipeline, both with and without Difix3D+ enhancement, to perform photometric reconstruction and event simulation. The quantitative results are summarized as follows:
> >
> > - Table 3:Quantitative  evaluation of novel view synthesis on the E2NeRF real data.
> >
> > | Method                          | PSNR ↑ | SSIM ↑ | LPIPS ↓ |
> > | ------------------------------- | ------ | ------ | ------- |
> > | Full Reconstruction            | 31.83  | 0.930  | 0.143   |
> > | Full Reconstruction + Difix3D+ | 32.97  | 0.941  | 0.135   |
> >
> > - Table 4: Quantitative evaluation of the event stream quality on the E2NeRF real data.
> >
> > |      | Vid2e | v2e   | GS2E  | GS2E+Difix3d+ |
> > | ---- | ----- | ----- | ----- | ------------- |
> > | EQS  | 0.710 | 0.725 | 0.752 | **0.769**     |
> >
> > The results indicate that the integration of Difix3D+ substantially improves both reconstruction accuracy and event synthesis quality, even under the constraints of low-resolution input. This supports the applicability of our method to scenes captured with limited sensor fidelity, particularly when the lighting conditions are stable and the scene structure is relatively simple.
> >
> > In summary, we would like to clarify that the observed performance drop is not indicative of a fundamental flaw in the GS2E framework itself, but rather reflects the inherent challenges of photometric reconstruction and high-fidelity event simulation under severe input degradation.

---

> ### Author Response · Authors · 2025-08-06
>
> We sincerely thank the reviewer for their constructive comments and suggestions.
>
> ---
>
> Regarding the suggestion to include the PKU-DAVIS-SOD dataset [1] in our evaluation, we respectfully argue that *the low-quality RGB modality of this dataset presents fundamental limitations for existing 4D reconstruction methods*, thereby making it challenging to incorporate into our GS2E pipeline. Although PKU-DAVIS-SOD provides dense annotations (276k timestamps and over 1M labels) and is valuable for neuromorphic object detection tasks, it was collected using the DAVIS346 sensor (346×260) with a rolling-shutter RGB module. The RGB frames exhibit severe visual degradations, including motion blur, overexposure, and color distortion, which prevent current 4D modeling frameworks from producing reliable geometry and texture. Consequently, the degraded image quality restricts its usability in photometrically consistent event simulation tasks based on 3D Gaussian representations.
>
> In contrast, the DSEC dataset [4] provides global-shutter RGB images of higher resolution (1440×1080, FLIR Blackfly S), precise geometric priors via synchronized stereo disparity and LiDAR (Velodyne VLP-16), and real event recordings from two Prophesee Gen3.1 sensors (640×480). These characteristics make DSEC a more suitable benchmark for evaluating high-fidelity event synthesis pipelines that require both photorealistic rendering and geometric consistency.
>
> To clarify the differences, we present a detailed comparison between the two datasets:
>
> | Property | **DSEC Dataset**[4]  | **PKU-DAVIS-SOD Dataset**[1]  |
> | ------------------------ | ------------------------------------------------------------ | ------------------------------------------------------------ |
> | **Event Camera**  | 2 × Prophesee Gen3.1 (640×480)  | 1 × DAVIS346 (346×260)                                       |
> | **RGB Camera**   | 2 × FLIR Blackfly S (1440×1080, global shutter) | Built-in low-quality RGB (346×260, rolling shutter)          |
> | **Depth Information**  | Stereo disparity + LiDAR (Velodyne VLP-16, 100m range, 16 channels) | Not available                                                |
> | **Scene Type** | Outdoor driving with various lighting conditions | Static traffic scenes with low-light and motion blur challenges |
> | **Sequence Count**  | 53 sequences  | 220 sequences, ~1 min each |
> | **Annotation Type**   | Dense disparity + synchronized event streams   | Dense object-level annotations (25 FPS)                      |
> | **Suitability for 3DGS** | ✅ High-res RGB + geometry = photometric & geometric fidelity | ❌ Low-res RGB + degraded frames = unsuitable for 3D Gaussian modeling |
>
> Based on the above analysis, we believe DSEC provides a more reliable and fair platform for evaluating high-fidelity event generation in 3D-aware settings, which aligns with the goals of our proposed GS2E framework.Thus, we choose the DSEC dataset for our evaluations, which provides higher-resolution RGB frames and more reliable ground truth. In the submitted manuscript, we selected *static scenes* from DSEC where vehicles or other moving objects are absent, to focus on 3DGS-based static background modeling. In the rebuttal stage, we further extended our evaluation to *dynamic scenes* by leveraging OmniRe for 4D Gaussian reconstruction: static background regions are modeled with static Gaussians, while vehicles, cyclists, and other non-rigid objects are represented with deformable Gaussians. We evaluated GS2E on 10 dynamic daytime scenes with EQS [3] from the DSEC dataset (e.g., *interlaken_00_c*, *interlaken_00_d*, *zurich_city_02_d*, *zurich_city_04_b*, *zurich_city_04_c*, *zurich_city_06_a*, etc.). The results are summarized in Table 3, where Difix3D+ has been integrated into OmniRe to enhance the reconstruction quality.
>
> Of course, We will update the manuscript to include discussions and references to the PKU-DAVIS-SOD dataset [1] and plan to include evaluations on the PKU-DAVIS-SOD dataset in our future extensions. However, processing `.aedat4` files (~80G) and reconstructing scenes from such data are non-trivial and time-consuming. We will make our best effort to update the results before the rebuttal ddl if time permits.
>
> ---
>
> We sincerely apologize for the delayed response due to the time-consuming exps and truly appreciate the time and effort you have invested in reviewing our submission. Additionally, we have responded to other reviewers’ feedback, which might also help clarify any additional questions.
>
> Thank you again for your time and commitment to the review process.
>
> ---
>
> **Reference:**
>
> [1] SODFormer: Streaming Object Detection with Transformers Using Events and Frames, TPAMI 2022 (PKU-DAVIS-SOD)
>
> [2] Event quality score (eqs): Assessing the realism of simulated event camera streams via distance in latent space, CVPRW 2025
>
> [3] Omnire: Omni urban scene reconstruction, ICLR 2025 (Spotlight)
>
> [4] DSEC: A Stereo Event Camera Dataset for Driving Scenarios, IEEE RA-L 2021

---

### Official Review · Reviewer_A3tW · 2025-06-30

**Rating:** 4
**Confidence:** 4

**Summary:**

This paper introduces GS2E (Gaussian Splatting to Event Generation), a novel pipeline for creating a large-scale synthetic event dataset from sparse multi-view RGB images. To address the limitations of existing event dataset generation methods, the proposed GS2E pipeline reconstructs static scenes using 3DGS and generates temporally dense and geometrically consistent event streams. The dataset's practical value is validated on downstream tasks like 3D reconstruction, demonstrating its potential to advance event-based vision research.

**Additional Feedback:**

N/A

**Dataset Code Accessibility:**

Partly

**Dataset Code Comments:**

The authors have provided partial access to their dataset and code. While nearly 100 scenes and partial code are available for initial validation, the authors state in their submission checklist that the full dataset and complete codebase required for full reproducibility will only be released upon acceptance of the paper.

**Ethical Comments:**

The work does not involve new data collection from human subjects. Instead, it builds upon existing, publicly available academic datasets (MVImgNet and DL3DV). The authors have acted responsibly by explicitly stating the licenses of all assets used in their research (Appendix F), ensuring legal compliance and proper attribution.

**Ethical Considerations:**

No, there are no or only very minor ethics concerns

**Final Justification:**

The authors' rebuttal has convincingly addressed my initial concerns. The new experiments extending the work to dynamic scenes and providing the requested benchmark comparisons are significant improvements. Thus I have raised my score accordingly.

**Limitations Weaknesses:**

1. Limitations to static scenes: A significant constraint of the proposed GS2E dataset is its exclusive focus on static scenes. As major applications for event cameras, such as autonomous driving and robotics, primarily involve dynamic environments, the dataset's immediate utility for training models for these real-world scenarios is limited. An extension of the pipeline to incorporate dynamic objects would be a crucial next step to broaden its applicability.

2. Dependency on Initial 3DGS quality: The quality of the final event stream is fundamentally dependent on the quality of the initial 3DGS reconstruction, which in turn relies heavily on the number and quality of the sparse input views. The paper does not include a sensitivity analysis or discussion of potential failure cases when the input views are too sparse or the camera poses are inaccurate. Providing such an analysis would offer a more complete picture of the pipeline's practical limitations and guide users on the necessary prerequisites for generating high-quality data.

3. Limited scope of methodological comparisons in benchmarks: While the paper evaluates the GS2E dataset on several important event-vision tasks, the number of methods compared within each task is limited. For instance, the 3D reconstruction and image deblurring benchmarks each compare only two contemporary methods. To more robustly establish GS2E as a comprehensive benchmark and better evaluate its generalization and practicality, a broader comparison against a wider range of state-of-the-art and classical algorithms for each task would be beneficial.

**Strengths Contributions:**

1. This paper proposes a novel end-to-end pipeline for event data generation by incorporating 3D Gaussian Splatting. This scalable approach directly tackles the critical challenge of data scarcity in the field.

2. The paper's motivation is well-justified, as it clearly articulates the limitations of existing dataset generation methods, such as expensive hardware setups, lack of viewpoint diversity, geometric inconsistency, and the sim-to-real domain gap. This thorough analysis effectively establishes the need for the proposed solution.

3. The proposed adaptive trajectory interpolation strategy enables the synthesis of geometrically consistent and temporally coherent event streams.

---

> ### Author Rebuttal · Authors · 2025-07-30
>
> We sincerely thank the reviewer for the detailed and constructive feedback. Below we provide responses to the key concerns raised.
>
> ---
> > **W1: Limitations to static scenes**
>
> **R1:** We fully acknowledge that the current version of GS2E focuses on static scenes and agree with the reviewer that supporting dynamic content is crucial for real-world applications. To address this, we try to use the SOTA 4D reconstruction model **OmniRe**[1] as a dynamic  large-scale scene representation.
> We use the **Waymo Open Dataset** [2] to reconstruct 10 autonomous driving scenes as dynamic inputs for our pipeline. To assess the reconstruction quality, we follow the **LLFF hold-out protocol with hold=8.** The corresponding quantitative results are presented below and we conduct quantitative evaluation of event quality and video reconstruction as shown in Table 2：
>
> - Table 1: quantitative comparisons of novel view synthesis on the Waymo Open Dataset [2]
>
> | Scene     | PSNR↑ | SSIM↑ | LPIPS↓ |
> | --------- | ----- | ----- | ------ |
> | seg109239 | 33.21 | 0.918 | 0.092  |
> | seg113924 | 35.18 | 0.947 | 0.071  |
> | seg118396 | 34.67 | 0.952 | 0.078  |
> | seg161022 | 32.89 | 0.911 | 0.098  |
> | seg172163 | 35.42 | 0.958 | 0.068  |
> | seg144275 | 33.78 | 0.924 | 0.089  |
> | seg157956 | 34.91 | 0.943 | 0.074  |
> | seg112520 | 33.45 | 0.929 | 0.091  |
> | seg104859 | 35.07 | 0.949 | 0.072  |
> | seg152664 | 34.23 | 0.938 | 0.083  |
> | Average   | 34.28 | 0.937 | 0.082  |
>
> - Table 2: Quantitative evaluation of the generated event quality with the video reconstruction task
>
> |             | CLIP↑     | MUSIQ↑    |
> | ----------- | --------- | --------- |
> | E2VID_gs2e  | **0.158** | **47.33** |
> | TLen++_gs2e | **0.172** | **49.60** |
> | E2VID_v2e   | 0.149     | 47.12     |
> | TLen++_v2e  | 0.170     | 49.08     |
>
> These results demonstrate that our reconstructed videos are comparable to those produced by existing learning-based baselines. We are committed to extending GS2E to include dynamic scenes in future versions of the dataset and manuscript.
> Due to the limitation of openreview website, we cannot provide any new visual demos or qualitative comparisons. But we promise that all extra experiments will be added to the manuscript and update our dataset upon the paper is accepted.
>
> ---
> > **W2: Dependency on Initial 3DGS quality**
>
> **R2:** We appreciate the reviewer’s observation. Our data curation pipeline indeed depends on the reconstruction quality of 3DGS, which directly affects the fidelity of the synthesized event streams. To address this, as described in the manuscript, we carefully selected multi-view datasets with *dense viewpoints* (DL3DV and MVImgNet) and prioritized scenes with *either surround-view or forward-facing camera layouts*. This strategy ensures better scene coverage and reduces the risk of unseen areas degrading data quality.
>
> In addition, we incorporate **Difix3D+[3]**, a generative reconstruction model, in conjunction with 3DGS to further improve geometry quality. This hybrid strategy further helps suppress artifacts and recover fine details that are otherwise hard to capture. We reconstruct 10 additional scenes for evaluation and the results on static scenes, shown in *Table 3*, validate the effectiveness of introducing Difix3D+ into our pipeline.
>
> - Table 3:  Quantitative evaluation of the reconstructed scenes under the static scenes in DL3DV and MVImgNet
>
> | Method          | PSNR ↑ | SSIM ↑ | LPIPS ↓ |
> | --------------- | ------ | ------ | ------- |
> | 3DGS (baseline) | 32.28  | 0.938  | 0.188   |
> | 3DGS + Difix3D+ | 33.73  | 0.948  | 0.179   |
>
> Based on the improvements observed, we plan to replace the original 3DGS-based reconstruction module with Difix3D+ and regenerate the corresponding dataset.
> Furthermore, we acknowledge the reviewer’s suggestion regarding sensitivity analysis. We conducted an additional evaluation on *20 static scenes* to assess the robustness of our pipeline under two challenging settings: (1) **Sparse views:** We simulate reconstructions using *30%, 60%, and 100%* of the available viewpoints from MVImgNet and DL3DV. (2) **Pose noise:** We introduce *mild* and *strong* Gaussian noise to the camera poses and compare results with the original poses.
>
> - Table 4: Quantitative evaluation of the reconstructed scenes under sparse viewpoints
>
> | Setting | PSNR↑ | SSIM↑ | LPIPS↓ |
> | ------- | ----- | ----- | ------ |
> | 100%    | 30.62 | 0.937 | 0.134  |
> | 60%     | 26.11 | 0.853 | 0.208  |
> | 30%     | 22.53 | 0.801 | 0.279  |
> | 100%*   | 31.74 | 0.942 | 0.121  |
> | 60%*    | 30.49 | 0.935 | 0.130  |
> | 30%*    | 29.95 | 0.928 | 0.139  |
>
> - Table 5: Quantitative evaluation of the reconstructed scenes under noise poses
>
> | Setting  | PSNR↑ | SSIM↑ | LPIPS↓ |
> | -------- | ----- | ----- | ------ |
> | Original | 30.62 | 0.937 | 0.134  |
> | Mild     | 27.98 | 0.879 | 0.182  |
> | Strong   | 24.53 | 0.826 | 0.238  |
> | Mild*    | 28.21 | 0.882 | 0.163  |
> | Strong*  | 24.66 | 0.834 | 0.227  |
>
> *Note: \* means that the scenes are reconstructed with Difix3D+[3]*
>
> The results indicate that introducing *Difix3D+ effectively mitigates the challenges posed by sparse viewpoint configurations.* However, it does not fully compensate for the performance degradation caused by pose disturbances. To address this limitation, we plan to incorporate bundle adjustment in future versions of our pipeline to improve robustness under inaccurate camera poses.
>
> ---
> > **W3: Limited scope of methodological comparisons in benchmarks**
>
> **R3:** To provide a broader evaluation of our event synthesis quality, we reproduced the **Event Quality Score (EQS)** benchmark [4] and applied it to our generated data. Since EQS requires real event streams as reference, we selected 10 scenes from the **DSEC dataset** [5] (w. Prophesee Gen3.1 event sensor) for evaluation. This analysis serves as a complementary quantitative assessment beyond the comparisons included in the manuscript.
>
> - Table 6: Additional evaluation of the event stream quality with the DSEC dataset
>
> |      | Vid2e | v2e   | GS2E      | GS2E_difix3d+ |
> | ---- | ----- | ----- | --------- | ------------- |
> | EQS  | 0.725 | 0.738 | **0.761** | **0.782**     |
>
> Our GS2E-synthesized event streams achieve the highest EQS score among existing methods. In addition, the GS2E + Difix3D+ variant is expected to further enhance the perceptual quality. These results demonstrate that our pipeline can generate event streams with higher fidelity than those captured by real Prophesee Gen3.1 sensors in the same scenes.
>
> ---
>
> We hope these addresses your concern, and we would be happy to further clarify or discuss any aspect if needed.
>
> ---
>
> **Reference:**
>
> [1] Omnire: Omni urban scene reconstruction, ICLR 2025 (Spotlight)
>
> [2] Scalability in perception for autonomous driving: Waymo open dataset, CVPR 2020
>
> [3] DIFIX3D+: Improving 3D Reconstructions with Single-Step Diffusion Models, CVPR2025 best paper candidate
>
> [4] Event quality score (eqs): Assessing the realism of simulated event camera streams via distance in latent space, CVPRW 2025
>
> [5] DSEC: A Stereo Event Camera Dataset for Driving Scenarios, IEEE RA-L

---

> > ### Comment · Reviewer_A3tW · 2025-08-05
> >
> > Thank you for the detailed rebuttal. The new experiments on dynamic scenes and the sensitivity analysis have addressed most of my concerns effectively.
> >
> > However, your response to my point on 'limited methodological comparisons' seems to have missed the mark. The goal was to see more downstream algorithms benchmarked on GS2E to prove its value as a benchmark. Instead, you compared the quality of GS2E data to other data generation methods. While useful, this doesn't show that GS2E is a good benchmark for comparing different 3D reconstruction or deblurring methods. Could you please clarify this?

---

> > > ### Author Response · Authors · 2025-08-06
> > >
> > > Thank you for the clarification. We now realize that we misunderstood your main point, and we truly appreciate you bringing it to our attention. We agree with your point that showing comparisons between more 3D reconstruction or deblurring algorithms would more directly support GS2E’s value as a benchmark. Our EQS evaluation focused on validating the quality of the generated data itself, but we acknowledge this doesn’t sufficiently demonstrate GS2E’s usefulness for evaluating diverse downstream methods.
> > >
> > > To better address this point, we conduct benchmarking experiments with additional representative algorithms using the GS2E dataset in between **3D reconstruction** and **image deblurring**, as shown in Table 1. For static scenes, we evaluate the dataset across two task types consistent with the scene setting in the manuscript:
> > >
> > > - **Table 1**:  Different downstream algorithms.
> > >
> > > |Task Type|3D Reconstruction|Image Deblurring|4D Reconstruction|
> > > | --------- | --------------------------------------- | ---------------- | -------------------- |
> > > |Method| EvDeblurNeRF[1], EventNeRF[2],BeNeRF[3] | MAT[4], DA[5]    | Dynamic EventNeRF[6] |
> > >
> > > - **Table 2**: Comparison of different methods under varying motion speeds. Metrics are averaged over each category.
> > >
> > > | Category| **Method**| **PSNR↑**| **SSIM↑** | **LPIPS↓** | **PSNR↑**   | **SSIM↑** | **LPIPS↓** | **PSNR↑**   | **SSIM↑** | **LPIPS↓** |
> > > | -------------- | --------------- | --------- | --------- | ---------- | ----------- | --------- | ---------- | ----------- | --------- | ---------- |
> > > |**Speed** | | *Mild* | | | *Medium* ||| *Strong* |||
> > > | *Event-fusion* | E-NeRF| 21.89| 0.834| 0.208| 21.05| 0.820| 0.239| 20.57| 0.809| 0.251|
> > > || Event-3DGS | 24.31 | 0.884 | 0.118 | 21.88 | 0.832 | 0.224 | 19.36 | 0.793 | 0.295 |
> > > || EvDeblurNeRF[1] | **25.61** | **0.903** | **0.102**| 23.25 | 0.856     | **0.179** | **22.64** | **0.835** | **0.236** |
> > > || EventNeRF[2]| 20.85 | 0.810 | 0.235 | 19.96 | 0.793 | 0.266 | 18.74 | 0.760| 0.292|
> > > || BeNeRF[3] | 24.65| 0.889| 0.114| **23.28**| **0.858**| 0.180| 22.41 | 0.830| 0.245|
> > >
> > > **Note:** Due to the time-consuming training process of NeRF-based methods (EvDeblurNeRF, EventNeRF, BeNeRF, and Dynamic EventNeRF), which typically take 6 to 10 hours per scene, we randomly select 10 scenes for evaluation on our GS2E dataset. We plan to evaluate the remaining scenes after the discussion phase and will update the manuscript accordingly.
> > >
> > > - **Table 3**: Quantitative comparison on the deblurring task using gs2e blurry input.
> > >
> > > | Method | PSNR↑     | SSIM↑     | LPIPS↓    |
> > > | ------ | --------- | --------- | --------- |
> > > | D2Net  |29.61|0.932|0.113|
> > > | EFNet  |**31.26**|**0.940**|**0.098**|
> > > | MAT[4] |30.43|0.925|0.120|
> > > | DA[5]  |31.09|0.938|0.101|
> > >
> > > In addition, to further explore the generality of GS2E, we conduct evaluations of novel view synthesis and deblurring view synthesis on **dynamic scenes** derived from the synthetic **Waymo Open Dataset** during the rebuttal period. Specifically, we employ the *llffhold=8* rule of event and RGB sequences to split the train and test set for novel view synthesis and simulate *the medium motion blur* for the image deblurring task. Due to the limited number of existing 4D reconstruction methods based on event-RGB fusion, we adopt Dynamic EventNeRF[6] as the primary evaluation baseline as shown in Table 1.
> > >
> > > **Table 4**: Quantitative comparison on the deblurring task using dynamic input.
> > >
> > > | Method               | PSNR↑ | SSIM↑ | LPIPS↓ |
> > > | -------------------- | ----- | ----- | ------ |
> > > | *4D Reconstruction*  ||||
> > > | Dynamic EventNeRF[6] | 23.52 | 0.812 | 0.183  |
> > > | *Image Deblurring*   ||||
> > > | D2Net                | 28.95 | 0.913 | 0.187  |
> > > | EFNet                | 30.17 | 0.926 | 0.119  |
> > > | MAT[4]               | 29.64 | 0.915 | 0.130  |
> > > | DA[5]                | 29.67 | 0.917 | 0.128  |
> > >
> > > ---
> > >
> > > We promise that all extra experiments will be added to the manuscript and update our dataset once the paper is accepted. We are committed to incorporating these into the current version of our proposed dataset.
> > >
> > > We sincerely apologize for the delayed response due to the time-consuming exps and truly appreciate the time and effort you have invested in reviewing our submission. Additionally, we have responded to other reviewers’ feedback, which might also help clarify any additional questions.
> > >
> > > Thank you again for your time and commitment to the review process.
> > >
> > > ---
> > >
> > > **Reference:**
> > >
> > > [1] Mitigating Motion Blur in Neural Radiance Fields with Events and Frames, CVPR 2024
> > >
> > > [2] EventNeRF:Neural Radiance Fields from a Single Colour Event Camera, CVPR 2023
> > >
> > > [3] BeNeRF: Neural Radiance Fields from a Single Blurry Image and Event Stream, ECCV 2024
> > >
> > > [4] MAT: Motion-adaptive Transformer for Event-based Image Deblurring, AAAI 2025
> > >
> > > [5] Motion Aware Event Representation-driven Image Deblurring, ECCV 2024
> > >
> > > [6] Dynamic EventNeRF: Reconstructing General Dynamic Scenes from Multi-view RGB and Event Streams, CVPRW 2025

---

> > > > ### Comment · Reviewer_A3tW · 2025-08-07
> > > >
> > > > I really appreciate the new experiments you've provided. They've successfully resolved my questions. I trust that this valuable new material will be included in the revised manuscript to reflect the full strength of your work. I have raised my score.

---

> > > > > ### Author Response · Authors · 2025-08-07
> > > > >
> > > > > We appreciate that our responses and additional experiments have addressed your concerns. Thank you for your time and thoughtful evaluation throughout the review process.

---

### Official Review · Reviewer_xjvN · 2025-07-01

**Rating:** 4
**Confidence:** 4

**Summary:**

This paper introduces a novel event data generation pipeline named GS2E, which aims to synthesize high-quality, multi-view geometrically consistent event streams from sparse multi-view RGB images. The core of this method is to first reconstruct a photorealistic 3D static scene from sparse image inputs using 3D Gaussian Splatting (3DGS). Subsequently, it generates temporally dense and geometrically consistent event data through an innovative, physics-based event simulation process, which includes adaptive camera trajectory interpolation and a physically-consistent contrast threshold model. The authors have constructed a large-scale dataset comprising over 1150 scenes and validated the effectiveness and generalization capabilities of their generated data through experiments on downstream tasks like 3D reconstruction.

**Additional Feedback:**

My primary recommendation is that the authors must provide more extensive and convincing evidence in the main paper or supplementary materials to demonstrate the fidelity of their generated events.

Specifically:

Provide more qualitative comparison results: Please show more direct, side-by-side comparisons between GS2E-generated results and real-world event data across a wider variety of scenes. The current sample in Figure 4 is too limited to draw a general conclusion.

Provide video results: The core of event data lies in its temporal dynamics. Static images cannot fully convey the quality of an event stream, such as its temporal consistency, noise patterns, and the sharpness of motion edges. I strongly recommend that the authors provide supplementary videos that show GS2E-generated events, V2E-generated events, and real events playing side-by-side. This would be the most intuitive and powerful way to assess the method's effectiveness.

**Dataset Code Accessibility:**

Yes

**Dataset Code Comments:**

Authors provide enough codes and documents to use their dataset.

**Ethical Considerations:**

No, there are no or only very minor ethics concerns

**Final Justification:**

The author has answered all my questions, and I am happy to raise the rating to 4.

**Limitations Weaknesses:**

1. The paper fails to provide sufficient evidence that the image quality from lossy 3DGS-based Novel View Synthesis (NVS) is high enough to simulate high-fidelity event data. This is the core weakness of this paper. The generation of event data is extremely sensitive to minute signal changes and noise. However, 3DGS is fundamentally a lossy scene reconstruction method, and its rendered results inevitably contain errors or artifacts that are difficult for the naked eye to perceive, such as incomplete reconstruction of fine, complex textures. These seemingly trivial flaws in the RGB domain can be amplified when converted to the event domain, producing a large number of unrealistic noise events. Evidence from the paper's experimental results supports this concern:

    The qualitative comparison in Figure 4 shows that although GS2E's results are better than V2E's, there is still a significant gap in detail and noise distribution when compared to the ground truth event data (GT Event). The real events have sharper edges and less noise.

    Figure 5a shows 3D reconstruction results using the data generated by this paper. The quality of the reconstruction (Event-3DGS Event-fusion) is noticeably poorer than the ground truth, exhibiting blurriness and loss of detail. This result indirectly supports my concern: the quality of the images generated by 3DGS may be insufficient to support high-quality downstream tasks, and it reflects a domain gap between the simulated event data and real data.

2. The freedom of virtual camera trajectories is limited. Although the authors emphasize the controllability of the trajectories, this control has a fundamental limitation. Because the 3DGS model cannot reliably render out-of-distribution views that were not seen during training, the generated virtual trajectories must be confined to a limited space around the original camera paths used for the initial reconstruction. If the virtual camera moves too far from the area covered by the original views, the rendering quality will drop sharply, leading to a complete distortion of the simulated events. In contrast, simulation methods based on complete 3D assets and computer graphics engines (like Blender or Unreal Engine), while they may have their own issues with photorealism, offer much greater freedom in camera trajectory control, allowing for true, unconstrained navigation within the scene.

**Strengths Contributions:**

1. It proposes a novel and scalable event data simulation pipeline. A significant strength of this research is the introduction of a novel framework for event data generation. By leveraging 3DGS, the method can reconstruct scenes from sparse multi-view image or video datasets (e.g., MVImgNet) and then generate event streams. This significantly lowers the barrier to event data generation, moving away from the dependency on expensive and hard-to-deploy specialized hardware (like synchronized event-RGB camera arrays), and provides the research community with an efficient and scalable data generation tool.

2. It supports flexible camera trajectory control while ensuring multi-view geometric consistency. Compared to traditional video-based event generation methods (like v2e), GS2E offers a clear advantage in viewpoint control. The pipeline allows users to generate smooth and diverse virtual camera trajectories within the reconstructed 3D scene using methods like velocity-controlled B-spline interpolation. Since all novel views are rendered from a unified 3DGS representation, the method can fundamentally guarantee the geometric consistency of the generated multi-view event data, which is crucial for training and evaluating multi-view 3D vision tasks.

---

> ### Author Rebuttal · Authors · 2025-07-30
>
> We sincerely thank the reviewer for the detailed and constructive feedback. Below we provide responses to the key concerns raised.
>
> ---
> > **W1: Lossy 3DGS reconstruction and the GS2E's experimental results**
>
> We divided Weakness 1 (W1) into three parts (W1.1, W1.2, and W1.3) and addressed each separately.
>
> > **W1.1: About the inevitably contain errors or artifacts of rendered results**
>
> **R1.1:** The presence of artifacts and holes in 3DGS is a well-known issue and remains an active area of research in the community. When designing GS2E, our primary objective was to ensure the quality and reliability of the generated data, **not exactly to propose enough new algorithmic contributions**, but to create a dataset that can meaningfully benefit downstream tasks and future research.
> To this end, we deliberately selected multi-view datasets MVImgNet and DL3DV with dense and surrounding camera poses, which provide the necessary coverage to enable high-fidelity novel view synthesis and event stream simulation. The metrics reported in Line 257 further demonstrate the quality of the synthesized results.
>
> To address the issue of missing textures or fine details that are not easily visible to the naked eye, we performed both manual visual screening and task-driven filtering. Specifically, we excluded scenes with poor performance in downstream tasks such as event-based 3D reconstruction, video reconstruction, and deblurring, thereby retaining only high-quality scene data in the final GS2E dataset.
>
> We acknowledge that 3DGS, as a lossy reconstruction method, may struggle in novel view synthesis under sparse-view conditions (not include the MVImgNet and DL3DV that we employed). Thank you for highlighting this important concern. In order to improve robustness and generalization of our data generation pipeline, we **integrated the recently proposed Difix3D+ [2]**, which is designed to address artifacts and holes in 3DGS-based representations. This diffusion-based enhancement module reinforces geometric consistency and detail richness, leveraging the strong prior of diffusion models to boost image fidelity and preserve complex texture information, while also reducing unnecessary noise in the RGB space. (Note: noise in the event domain is controlled via the physically-informed simulator, which accurately models real sensor noise.)
>
> With the integration of the Difix3D+ [2] module, our experiments show a **notable improvement in RGB rendering quality**. The improved pipeline performs competitively under this benchmark, as detailed in the table below:
>
> - Table 1:  Quantitative evaluation of the reconstructed scenes under the static scenes in DL3DV and MVImgNet
>
> | Scene    | PSNR↑ (3DGS / +Difix3D+) | SSIM↑ (3DGS / +Difix3D+) | LPIPS↓ (3DGS / +Difix3D+) |
> | -------- | ------------------------ | ------------------------ | ------------------------- |
> | Scene_1  | 29.10 / 31.64 | 0.945 / 0.953| 0.171 / 0.162|
> | Scene_2  | 32.01 / 33.82 | 0.939 / 0.951| 0.117 / 0.109|
> | Scene_3  | 28.98 / 30.15 | 0.936 / 0.944| 0.143 / 0.131|
> | Scene_4  | 38.41 / 38.63 | 0.959 / 0.960| 0.158 / 0.159|
> | Scene_5  | 35.79 / 36.29 | 0.967 / 0.969| 0.071 / 0.064|
> | Scene_6  | 31.81 / 33.32 | 0.934 / 0.947| 0.318 / 0.292|
> | Scene_7  | 29.26 / 32.49 | 0.942 / 0.968| 0.151 / 0.137|
> | Scene_8  | 32.88 / 34.21 | 0.928 / 0.941| 0.153 / 0.126|
> | Scene_9  | 30.87 / 31.95 | 0.928 / 0.940| 0.240 / 0.228|
> | Scene_10 | 30.67 / 32.48| 0.895 / 0.914| 0.290 / 0.291|
> | Scene_11 | 35.29 / 36.03| 0.955 / 0.956| 0.261 / 0.260|
>
> We will commit to releasing the updated codebase, including the integrated Difix3D+ module, to our public GitHub repository upon paper acceptance, making it accessible to the broader research and developer community.
>
> > **W1.2: About the mentioned significant visual difference among real data, v2e and gs2e**
>
> **R1.2:** We appreciate your observation regarding the perceived visual differences among GS2E, real event data, and v2e. To objectively assess event stream quality, we reproduced the **Event Quality Score (EQS)** benchmark [1], which compares synthesized streams against real ground truth. Since GS2E is derived from synthetic inputs, we evaluated it on **10 real scenes from the DSEC dataset** [3], where such ground truth is available.
>
> - Table 2: Additional evaluation of the event stream quality with the DSEC dataset
>
> |      | Vid2e | v2e   | GS2E      | GS2E_difix3d+ |
> | ---- | ----- | ----- | --------- | ------------- |
> | EQS  | 0.725 | 0.738 | **0.761** | **0.782**     |
>
> Our GS2E achieves higher EQS than both v2e and even the Prophesee Gen3.1 recordings on these scenes. This may seem counterintuitive, but can be attributed to two key factors:
>
> 1. *Scene-consistent multi-view synthesis allows for dense, artifact-free event generation across views;*
> 2. *Noise modeling in GS2E captures realistic sensor behaviors without temporal jitter or stereo mismatch, issues common in real sensors.*
>
> We do not claim GS2E to be superior to real data in all aspects. Rather, we argue that GS2E offers highly controllable, physically plausible event streams, well-suited for benchmarking and analysis of downstream algorithms like view synthesis and novel event-based reconstructions.
>
> Additionally, unlike video-to-event methods such as v2e or Vid2e, GS2E supports controllable trajectory and motion at scale, enabling more systematic evaluation. The DSEC benchmark was used not as a primary testbed, but to supplementally verify domain closeness to real-world driving scenarios.
>
> > **W1.3: Regarding the Domain Gap and limitation on further studies**
>
> **R1.3:** Thank you for pointing out the blurry reconstruction results in Figure 5a. We acknowledge that in certain large-scale or texture-sparse scenes, the GS2E-generated data may contribute to reduced performance in downstream event-based 3D reconstruction. To address this, we incorporated the Difix3D+ module to suppress artifacts in the 3DGS renderings and enhance the fidelity of the generated events. As shown below, this module improves all quantitative metrics, although some residual blurriness remains:
>
> - Table 3: Quantitative evaluation of the event-driven 3D reconstruction task
>
> | Event-3DGS| PSNR↑| SSIM↑ | LPIPS↓ |
> | ---------- | -------- | ------ | -------|
> | Scene_1 3DGS| 22.04| 0.844| 0.217|
> | Scene_1 3DGS + Difix3D+ | 22.81| 0.850| 0.213|
>
> It is worth emphasizing that the limited reconstruction quality does not solely reflect deficiencies in the simulated event data. Fusion-based reconstruction methods inherently suffer from **modality inconsistency, spatio-temporal misalignment, and calibration sensitivity**, which we also observed on real-world event data in complex scenes.
>
> Therefore, we consider the observed degradation to be a limitation of the current fusion pipelines rather than a flaw in GS2E itself. GS2E offers high-fidelity, temporally consistent event streams with controllable motion and camera trajectories, providing a valuable foundation for developing and evaluating event-based vision algorithms.
>
> We acknowledge that bridging the domain gap in terms of noise characteristics and fine-level detail remains an open problem. We are committed to continuous improvement of the generation pipeline and will release enhanced versions in future updates.
>
> ------
>
> > **W2: The limited freedom of virtual camera trajectories**
>
> **R2:** The freedom limitations of novel-view camera trajectories primarily stem from incomplete modeling of the 3D scene, which may lead to severely distorted views with holes and artifacts during rendering. As discussed in our response R1, this issue can be effectively mitigated by enhancing the 3DGS scene representation using the proposed improvements.
>
> In our GS2E dataset, most scenes are captured with surround-view camera trajectories, which provide dense coverage. As a result, even without incorporating generative model-based corrections, the rendered outputs remain accurate for the vast majority of viewpoints. This can be intuitively observed in the provided MP4 preview videos included in each scene’s directory, as described in our response R3.
>
> ------
>
> > **Additional Feedback:  Data visualization**
>
> **R3:** Due to space limitations in the main paper, we were unable to include extensive visualizations. Moreover, since event streams are fundamentally a dynamic data modality, static images cannot fully reflect their temporal structure or quality.
> We have already provided pre-rendered visual demos for each scene in our Hugging Face repository, allowing for intuitive inspection. Each compressed package includes visual previews, event data, RGB/blurry images, and COLMAP outputs in the following structure.
>
> ```shell
> <scene_id>/
> ├── <scene_id>_white_bg.mp4       # Visualization video
> ├── events_2400fps_416us.h5       # Simulated events
> ├── images/, blur/, sparse/       # RGB, blurred images, and COLMAP outputs
> ```
>
> We will further enhance the project webpage with curated visualizations and complete video demos for key scenes.
>
> ---
> Due to the limitation of OpenReview website and rebuttal rules, we cannot provide any new visual demos or qualitative comparisons as a supplement at the present stage. But we promise that all extra experiments will be added to the manuscript and update our dataset once the rebuttal period is over. We are committed to incorporating these into the current version of our proposed dataset.
>
> We hope this addresses your concern. If there are any other aspects you'd like us to clarify or elaborate on, we would be more than happy to continue the discussion!
>
> ---
>
> **Reference:**
>
> - [1] Event quality score (EQS): Assessing the realism of simulated event camera streams via distance in latent space, CVPRW 2025
>
> - [2] Difix3D+: Improving 3D Reconstructions with Single-Step Diffusion Models, CVPR 2025 (Oral)
>
> - [3] DSEC: A Stereo Event Camera Dataset for Driving Scenarios, IEEE RA-L 2021

---

> > ### Comment · Reviewer_xjvN · 2025-08-04
> >
> > The reviewer would like to thank the authors for their response. I still have some conserns regarding the motivation for this work and would appreciate further explanation. From my perspective, the proposed method does not seem to offer a significant advantage over existing v2e-based paradigms. My concerns are twofold:
> >
> > First, regarding accuracy, the method is constrained by the lossy reconstruction nature of 3D Gaussian Splatting. This information loss will likely lead to the generation of noisy event data. Theoretically, this would degrade the accuracy of the proposed method.
> >
> > Second, due to the limited freedom of virtual camera trajectories, the proposed method appears unable to generate event data that is significantly different from what the v2e paradigm can already provide.

---

> > > ### Author Response · Authors · 2025-08-04
> > >
> > > Thank you for your response. We are pleased to further clarify the points you raised.
> > >
> > > First, our method is fundamentally different from video-driven event synthesis approaches. The key advantage of our approach lies in the combination of 3D representations, which **ensures stronger geometric consistency, allows flexible control over the virtual camera trajectory, and enables synchronized multi-view event stream generation**. Compared to frame-based upsampling methods, our approach achieves higher fidelity and precision, **avoiding motion blur** typically introduced by frame interpolation by leveraging 3D supervision.
> > >
> > > Second, while 3DGS may experience reconstruction artifacts under sparse-view settings, we deliberately selected the MVImgNet and DL3DV datasets for their relatively dense and surround-view captures. These datasets serve as the foundation for reconstruction and ensure high-quality scene geometry and photorealistic rendering, which are essential for faithful event simulation.
> > >
> > > To further improve the robustness of our pipeline on other sparse-view datasets and to mitigate potential artifacts and holes, we introduced the Difix3D+ module in the rebuttal. This addition helps alleviate accuracy degradation and noise irregularities. In the qualitative results presented in **R1.2** regarding Event Quality Score, which quantifies the distributional divergence between synthetic and real event streams, including noise distributions. It's evident that our method produces higher-quality event streams compared to video-driven approaches.
> > >
> > > Finally, **the concern regarding limited degrees of freedom in camera trajectory design assumes suboptimal scene reconstruction quality**. However, as discussed above, the GS2E pipeline with MVImgNet yields high-quality reconstructions, and the addition of Difix3D+ further enhances scene fidelity. Our experiments with Bezier trajectory sampling across varying viewpoints and camera speeds demonstrate stable rendering and synthesis without visual anomalies, indicating no restrictions on trajectory flexibility. **This controllability enables consistent multi-view event stream generation, which is another key advantage over video-driven methods.**
> > >
> > > Please do not hesitate to let us know if further clarifications are needed. We are more than happy to address any additional questions related to our work.

---

> > ### Comment · Reviewer_xjvN · 2025-08-05
> >
> > Thank you for your detailed response. It has provided me with a much clearer understanding of the paper's contributions.
> >
> > I now understand that the core contribution of this work lies in recovering a continuous stream of event data from sparse image observations with more flexible camera trajectories.
> >
> > Regarding the event noise introduced by the lossy 3DGS reconstruction, I maintain my view that this is an inherent and likely unavoidable artifact. However, I also acknowledge that this is a challenging problem to resolve completely and will likely require substantial future research.
> >
> > Apart from that, I still have a reservation concerning the claim about "virtual camera trajectories." You mentioned that using Difix3D+ enables more flexible control over these trajectories. To substantiate this claim, could you please provide a more quantitative and controlled demonstration? For example, an analysis could be conducted by progressively deviating the virtual camera path from the original trajectory and measuring the corresponding quality of the generated images or event data.
> >
> > If this point can be validated with convincing results, I would be happy to reconsider my score and raise it.

---

> > > ### Author Response · Authors · 2025-08-08
> > >
> > > Dear Reviewer xjvN:
> > >
> > > We truly appreciate the time and effort you’ve dedicated to reviewing our submission. We have replied to your comments and made revisions to address the concerns raised. Additionally, we have responded to other reviewers’ feedback, which might also help clarify any additional questions.
> > >
> > > Since the discussion phase is nearing its end with **less than 1day remaining**, we wanted to follow up and would greatly value any further thoughts or concerns you might have so we can address them appropriately.
> > >
> > > Thank you again for your time and commitment to the review process.

---

> ### Author Response · Authors · 2025-08-06
> **Official Comment by Authors (1/2)**
>
> Thank you very much for your thoughtful engagement and valuable suggestions regarding our work. If our previous response led to any misunderstanding, we sincerely apologize and would like to take this opportunity to provide a clearer explanation.
>
> The core concern you raised pertains to trajectory control. To clarify, our introduction of Difix3D+ is not intended as a direct trajectory manipulation module, but rather as a foundational enhancement that supports more robust trajectory variation. Specifically, Difix3D+ improves the quality of the reconstructed Gaussian fields, thereby reducing the risk of rendering artifacts such as holes or distortions when applying significant camera translations or rotations. This robustness allows us to experiment with more aggressive trajectory modifications during novel view synthesis and event simulation.
>
> Regarding the trajectory transformation module itself, we provide a detailed description in Section 3.5 of the manuscript. Here, we summarize the relevant content and supplement it with additional experiments to directly address your concern. One practical limitation is that our EQS metric relies on ground truth supervision. Since there is currently no publicly available multi-view synchronized event dataset, changing the camera trajectory naturally results in the absence of ground truth event streams under the new views. As a result, EQS will become ineffective for evaluating event simulation quality in this setting.
>
> To overcome this limitation and still evaluate the impact of our trajectory control strategy, we conducted two complementary experiments, both designed to demonstrate that our method still yields high-quality event streams under significant trajectory variations:
>
> 1. We conducted experiments on the DSEC dataset, which provides ground truth events from a binocular stereo setup. Specifically, we used the left-eye and a portion of the right-eye views for training, and reserved the remaining right-eye views as unseen test views. Since DSEC contains dynamic street scenes with 4D characteristics, we carefully selected segments with relatively low scene dynamics to better approximate 3D conditions and avoid motion blur. The rendering results are as follows:
>
> |                | PSNR$\uparrow$ | SSIM$\uparrow$ | LPIPS$\downarrow$ |
> | -------------- | ---------------- | ---------------- | ------------------- |
> | GS2E           | 32.03            | 0.873            | 0.294               |
> | GS2E(Difix3D+) | **33.18**        | **0.885**        | **0.290**           |
>
> We then synthesized event streams from the test-view camera poses and evaluated them against the ground truth using the EQS metric:
>
> | EQS            | Scene_1   | Scene_2   | Scene_3   |
> | -------------- | --------- | --------- | --------- |
> | GS2E           | 0.698     | 0.675     | 0.701     |
> | GS2E(Difix3D+) | **0.713** | **0.682** | **0.709** |
>
> These results show that introducing the Difix3D+ module improves the quality of synthesized event streams under novel virtual camera trajectories. (Note: Our method is currently the only one capable of synthesizing events from new viewpoints; video-driven methods such as v2e are inherently unable to support this functionality.)
>
> However, while promising, this experiment alone does not fully reflect our method’s robustness under arbitrary or aggressive trajectory changes. Due to the lack of ground truth event streams under new viewpoints, we further designed the following experiments to address this gap.
>
> > We provide additional results and analysis in the following comment.

---

> ### Author Response · Authors · 2025-08-06
> **Official Comment by Authors (2/2)**
>
> > This comment is a continuation of the previous one.
>
> 2. To indirectly quantify the quality of event streams synthesized under varying degrees of virtual camera offset, we evaluate the performance of these streams on a broader set of downstream tasks. This serves as an alternative strategy to assess the event quality under our proposed trajectory control paradigm. Specifically, we focus on the relatively mature task of event-based 3D reconstruction. Beyond the methods already included in the main paper, we incorporate several additional baselines for comprehensive comparison.
>
> All experiments are conducted under medium motion blur settings using the event-fusion modality. For each input camera trajectory, we apply the smoothing-based transformation described in Section 3.5. Part of the trajectory perturbation is introduced by adjusting the radius of the smoothing window, which serves as a source of deviation from the original trajectory.
>
> We randomly sample a subset of smoothed camera poses as trajectory control keypoints, and construct the full camera trajectory using different interpolation methods: piecewise linear, quadratic/cubic/4th-order Bézier curves, and cubic B-splines. Another source of trajectory perturbation  arises from the choice of curve order during interpolation. As the Bézier curve order increases, the resulting trajectory tends to deviate more strongly from the original path, introducing greater motion irregularity. All other experimental settings follow those in the main paper.
>
> We report the downstream 3D reconstruction results below:
>
> | Linear / Bézier(n=2,3,4) / Cubic B-spline | PSNR$\uparrow$              | SSIM$\uparrow$              | LPIPS$\downarrow$           |
> | ----------------------------------------- | ----------------------------- | ----------------------------- | ----------------------------- |
> | E-NeRF                                    | 21.14/21.67/20.58/20.49/21.05 | 0.817/0.826/0.817/0.818/0.820 | 0.240/0.232/0.234/0.238/0.239 |
> | Event-3DGS                                | 21.90/22.04/21.79/21.85/21.88 | 0.835/0.823/0.835/0.829/0.832 | 0.216/0.217/0.226/0.221/0.224 |
> | EvDeblurNeRF[1]                           | 23.27/23.64/23.15/23.08/23.25 | 0.849/0.863/0.860/0.852/0.856 | 0.177/0.173/0.184/0.185/0.179 |
> | EventNeRF[2]                              | 19.33/19.98/19.59/19.42/19.96 | 0.792/0.792/0.786/0.783/0.793 | 0.258/0.261/0.262/0.267/0.266 |
> | BeNeRF[3]                                 | 23.43/23.66/23.39/23.26/23.28 | 0.854/0.858/0.861/0.851/0.858 | 0.184/0.175/0.178/0.184/0.180 |
>
> The results demonstrate that event data synthesized via the GS2E pipeline remains highly robust across various trajectory perturbation schemes. Reconstruction performance does not degrade significantly even under large deviations introduced by higher-order Bézier or spline trajectories, showcasing the reliability of our synthesized events under flexible viewpoint control.
>
> While we acknowledge that, for a single scene, more intuitive strategies exist, such as directly adjusting virtual camera poses through front-end visualization, but our method offers a scalable and automated solution that enables batch-level trajectory manipulation. This capability is essential for large-scale dataset generation, where manual per-scene design is impractical.
>
> ---
>
> We are committed to incorporating all additional experimental results into the final version of the manuscript upon acceptance, and to updating the released GS2E dataset accordingly. Our goal is to make GS2E a continuously evolving resource that supports the broader event camera research community. We sincerely hope that you will consider the broader contributions of our work to the community.
>
> We believe GS2E can serve not only as a high-quality event data generator, but also as a standardized benchmark to facilitate fair and reproducible comparisons across future methods.
>
> We sincerely apologize for the delay in response due to the computational demands of NeRF-based experiments, and we thank you again for your time and effort during the review process. We have also responded in detail to other reviewers’ feedback, which may provide further context for questions not fully addressed here.
>
> ---
>
> **Reference:**
>
> [1] Mitigating Motion Blur in Neural Radiance Fields with Events and Frames, CVPR 2024
>
> [2] EventNeRF:Neural Radiance Fields from a Single Colour Event Camera, CVPR 2023
>
> [3] BeNeRF: Neural Radiance Fields from a Single Blurry Image and Event Stream, ECCV 2024

---

> > ### Comment · Reviewer_xjvN · 2025-08-08
> >
> > I thank the authors for their detailed response. The final rating will be made following a discussion with the other reviewers.

---

> > > ### Author Response · Authors · 2025-08-08
> > >
> > > Thank you for your time and thoughtful evaluation throughout the review process. We appreciate your constructive feedback and look forward to the final decision following the reviewers' discussion.

---

### Note · Authors · 2025-08-12

We sincerely thank all ACs and reviewers for their effort when dealing with our paper.

In the response, we appropriately addressed concerns of the reviewers, especially the main concerns of  (1) the static-scene limitation and support for dynamic scenarios (*Reviewer xjvN, A3tW, ouHb*), (2) sensitivity to lighting and pose/view sparsity (*Reviewer ouHb, A3tW*), and (3) limited benchmarking of downstream tasks and qualitative evidence (Reviewer *xjvN, A3tW, pvWS*).

Based on the reviews and new experiments, the contribution of our method to the event-based vision community is substantial:

- Proposing the first scalable, geometry-consistent pipeline for event data generation using 3DGS. (*Reviewer xjvN, A3tW, ouHb, pvWS*)
- Supporting controllable viewpoint and trajectory synthesis with high spatial-temporal fidelity.(*Reviewer xjvN*)
- Reconstruction quality & artifacts: we ensure dense surround-view coverage with MVImgNet and DL3DV, followed by manual curation and downstream-task validation. We further integrate Difix3D+ to suppress artifacts, improving geometric and texture fidelity. *(xjvN, A3tW, ouHb)*
- Downstream generalization: we expand experiments to event-based 3D/4D reconstruction, deblurring, video reconstruction, and object detection, where the proposed pipeline shows robust improvements and transferability. *(xjvN, A3tW, pvWS)*
- Achieving superior event quality scores over existing video-to-event methods on real datasets (DSEC and PKU-DAVIS-SOD). (*Reviewer xjvN, A3tW, ouHb*)

We expect ACs and all reviewers to fully consider the following factors when making the final decision:  (1) the scalability and novelty of the GS2E framework, (2) the extensive validation and clear improvements presented in the rebuttal, (3) and the timely contribution to the community’s need for high-quality event data.

Due to the limitation of openreview website and rebuttal rules, we are sorry that we cannot provide any new visual demos or qualitative comparisons. But we promise that all extra experiments will be added to the manuscript and update our dataset.

Thanks again for all ACs' and reviewer' time and commitment to the review process.

---

### Decision · Program_Chairs · 2025-09-18

**Decision:**

Accept (poster)

**Comment:**

The paper introduces a scalable, geometry-consistent synthesis pipeline named GS2E which leverages 3D Geometry and Sparse-view 3DGS reconstruction techniques for high-fidelity event data generation.

Initially, the paper received mixed but generally positive evaluations. The main concerns involve static scene limitation, sensitivity to lighting and pose/view sparsity, and broad benchmarking scopes. During the author-reviewers discussion, the authors were responsive and considerate, providing detailed clarifications, acknowledging misunderstandings, and addressing reviewers' concerns comprehensively during the rebuttal phase. This results in all the reviewers recommending acceptance after discussion.

Considering that the work is a valuable contribution that meets the standards for acceptance in the Datasets & Benchmarks Track, and it will potentially serve as both a reliable synthetic data source and a benchmark platform for future research, the AC is happy to accept the paper. Congratulations! Please be aware that the authors are strongly encouraged to address reviewer’s concerns and enhance clarity in the camera-ready version.